# A single atom change turns insulating saturated wires into molecular conductors

Xiaoping Chen[1,2], Bernhard Kretz [3], Francis Adoah[4], Cameron Nickle[4], Xiao Chi[5], Xiaojiang Yu [5], Enrique del Barco [4], Damien Thompson [6], David A. Egger[3✉] & Christian A. Nijhuis [1,2,7✉]

We present an efficient strategy to modulate tunnelling in molecular junctions by changing the tunnelling decay coefficient, $\beta$, by terminal-atom substitution which avoids altering the molecular backbone. By varying $X = H$, F, Cl, Br, I in junctions with $S(CH_2)_{(10-18)}X$, current densities ($J$) increase >4 orders of magnitude, creating molecular conductors via reduction of $\beta$ from 0.75 to 0.25 Å$^{-1}$. Impedance measurements show tripled dielectric constants ($\varepsilon_r$) with $X = I$, reduced HOMO-LUMO gaps and tunnelling-barrier heights, and 5-times reduced contact resistance. These effects alone cannot explain the large change in $\beta$. Density-functional theory shows highly localized, X-dependent potential drops at the $S(CH_2)_nX//$ electrode interface that modifies the tunnelling barrier shape. Commonly-used tunnelling models neglect localized potential drops and changes in $\varepsilon_r$. Here, we demonstrate experimentally that $\beta \propto 1/\sqrt{\varepsilon_r}$, suggesting highly-polarizable terminal-atoms act as charge traps and highlighting the need for new charge transport models that account for dielectric effects in molecular tunnelling junctions.

[1] Department of Chemistry, National University of Singapore, Singapore, Singapore. [2] Centre for Advanced 2D Materials and Graphene Research Centre, National University of Singapore, Singapore, Singapore. [3] Department of Physics, Technical University of Munich, Garching, Germany. [4] Department of Physics, University of Central Florida, Orlando, FL, USA. [5] Singapore Synchrotron Light Source, National University of Singapore, Singapore, Singapore. [6] Department of Physics, Bernal Institute, University of Limerick, Limerick, Ireland. [7] Hybrid Materials for Opto-Electronics Group, Department of Molecules and Materials, MESA+ Institute for Nanotechnology and Center for Brain-Inspired Nano Systems, Faculty of Science and Technology, University of Twente, P.O. Box 217, 7500 AE Enschede, The Netherlands. ✉email: david.egger@tum.de; c.a.nijhuis@utwente.nl

Significant effort has been dedicated to study and manipulate tunnelling rates across molecular wires, which serve as model systems to improve our understanding of the mechanisms of charge transport across molecules which, in turn, play a central role in, e.g., biological processes, catalysis, and energy conversion[1–4]. It is well-known that the tunnelling current density ($J$ in A/cm$^2$) decreases exponentially with the length of the molecular wire ($d$ in Å) given by the general tunnelling equation

$$J = J_0(V)e^{-\beta d} = J_0(V)10^{-\beta d/2.303}, \qquad (1)$$

where $J_0$ is a pre-exponential factor and the tunnelling decay coefficient ($\beta$ in Å$^{-1}$) determines how quickly the measured current decays with $d$[2,5–7]. In this context, unsaturated molecules with conjugated $\pi$-bonds are usually thought of as "molecular conductors" with low values of $\beta$ (0.1–0.4 Å$^{-1}$)[2,5,8,9] and saturated molecular wires with localized $\sigma$-bonds provide "molecular insulators" with large values of $\beta$ (0.8–1.2 Å$^{-1}$)[2,5,7,10,11]. This rule of thumb stands in sharp contrast with the high tunnelling rates established for various biomolecules[12,13], molecular wires of oligopeptides[14,15], and oligosilanes[16]. These all have saturated molecular backbones yet they exhibit low values of $\beta$ (0.1–0.5 Å$^{-1}$), and support long-range tunnelling over remarkably large distances of up to tens of nanometres[13,17].

So far, it has been challenging to engineer $\beta$ in experiments, and this difficulty is also reflected in various established mechanisms of charge transport across molecular wires. Often, coherent tunnelling is assumed (Eq. 1), where $\beta$ can be related to the tunnelling barrier height $\delta E_{ME}$ (defined by the offset in energy between the energy of the Fermi level, $E_F$, of the electrode and the energy of the molecular frontier orbital relevant for charge transport), as $\beta \propto \sqrt{\delta E_{ME}}$[2,5,18]. This explains why conjugated molecules, which often have frontier orbitals aligned close to $E_F$, have lower values of $\beta$ than saturated molecules, which have frontier orbitals further from $E_F$. Conversely, in the McConnell superexchange model, charge carriers tunnel via virtual states defined by the repeat units of the molecular wire; here, the tunnelling rate depends on the interaction strength between the repeat units of the molecular bridge[2,5,19–21]. This model has been used to explain low $\beta$ values (0.2–0.5 Å$^{-1}$) measured across tunnel junctions with self-assembled monolayers (SAMs) that have $\sigma$-bond backbones of oligoglycines[14], oligoprolines[22], and oligoglycols[15]. Furthermore, for very long molecules (e.g., proteins), a flickering resonance model has been proposed to explain long-range tunnelling and low $\beta$ values[13,23,24]. Finally, different types of hopping models have been proposed to explain low $\beta$ values of, for instance, bacterial nanowires[23], DNA[25], proteins[26], and long conjugated molecular wires[16,27,28]; here the value of $\beta$ also depends on the coupling strength between the repeat units, but these models predict a thermally activated component[27,28]. To summarize, all previous models suggest the necessity of tuning the chemical nature of the molecular wire to change the value of $\beta$.

We note that the value of $\beta$ also depends, besides the chemical nature of the molecular backbone[2,5,13–15], on the coupling strength between the molecules and electrodes ($\Gamma$) that is naturally related to $\delta E_{ME}$[28–30]. For molecular wires, where $\delta E_{ME}$ decreases with the number of repeat units due to an increase in conjugation with increasing molecular length, extremely low (<0.1 Å$^{-1}$)[27,28,31,32] and even negative $\beta$ values have been reported[31,33–35]. Such low $\beta$ values are also a signature of incoherent hopping and these junctions, in particular those containing redox centres, may operate in this hopping regime (also called incoherent tunnelling regime)[27,28,33]. Lambert and co-workers[36] were able to tune the $\beta$ value between 0.06 and 0.39 Å$^{-1}$ in Au–S(CH$_2$)$_n$FG(CH$_2$)$_n$S–Au junctions with a functional group FG = α-terthiophene, phenyl, or viologen. They found that changing the anchoring group from dithiol to

dithiolmethyl for FG = phenyl resulted in an increase of the $\beta$ value from 0.14 to 0.50 Å$^{-1}$ from which they concluded that localized states on the Au–S bond are involved in tunnelling along the FG units. In contrast, Frisbie and co-workers[37] found that $\beta$ values are similar for Au–S(CH$_2$)$_n$CH$_3$//Au and Au–S(CH$_2$)$_n$S–Au junctions, implying that localized states on the Au–S bond are not important for tuning $\beta$ (but note that they still significantly affect the contact resistance). Frisbie and co-workers[38,39] suggested that Stark effects are important to consider as they can cancel the potential effects of localized anchoring group-electrode states. Indeed, strong Au–S interaction results in severe broadening of the molecular states and therefore the Au–S states only occur as weak features in valence band spectra of aliphatic SAMs[38] (as also observed in the present study), highlighting the need to optimize the $\Gamma$ such that the molecular states remain localized in the molecule. Recently, Chen and co-workers[40] reported a method using bimetallic electrodes to enhance the conductance of HO$_2$C(CH$_2$)$_n$CO$_2$H single-molecule junctions via the surface $d$-band. They improved the interfacial interactions between molecules and transition metal electrodes, promoting interfacial electron transport. Here, we use junctions of the form Ag–S(CH$_2$)$_n$X//EGaIn ($n = 10$, 12, 14, 16, or 18, and X = H, F, Cl, Br, or I) where the weak interaction between the top electrode and the SAM allows us to investigate in detail how the terminal group X affects the tunnelling rates across the junctions.

So far, the influence of electrostatic effects in molecular tunnelling junctions on $\beta$ has been largely ignored. One way of quantifying trends in the electrostatics of various systems is by studying the static dielectric constant ($\varepsilon_r$) of molecular junctions, a macroscopic observable that can be measured via impedance spectroscopy[41]. Previous work focused mainly on $\pi$-conjugated systems and established that in densely packed SAMs, $\varepsilon_r$ hardly changes when the polarizability of the molecules of a SAM, $\alpha$, is tuned due to depolarization effects (e.g., induced dipoles in neighbouring molecules)[42–44]. However, it is not known how $\alpha$ affects the tunnelling behaviour of junctions in which depolarization effects are reduced to a minimum. In addition, the molecular ionization potential directly relates to $\alpha$ and, consequently, changing $\alpha$ affects molecular frontier orbital energies[45] and the energy level alignment of molecule–electrode interfaces[45,46], but it is disputed whether an increase in $\alpha$ changes the conductance of the junction[45–47]. Also not currently understood is how $\alpha$ affects the relationship between $\varepsilon_r$ and $\beta$. In principle, polarizable groups screen applied electric fields[48] or result in an induced dipole and, therefore, also affect the potential drop profile inside junctions[45,46]. Thus polarizable atoms or moieties are expected to have a large effect on the measured tunnelling rates, but so far experimental examples are rare and conflicting[45–47]. For instance, Whitesides and co-workers reported that the charge transport rates in metal-S(CH$_2$)$_n$FG//EGaIn junctions with aliphatic SAMs are independent of FG with FG being terminal aromatic groups[49], polar groups[50], ionic and/or hydrogen bonding groups[51], or halogen atoms[47], and concluded that changes in terminal group does not affect the charge transport rates. In these studies they used large junction areas of >1000 μm$^2$, but we have shown that such large junctions are prone to defects masking molecular effects and that, for EGaIn-based methods, stable junctions that are dominated by molecular effects should have an area of 300–500 μm$^2$ (ref. 52). Indeed, the Whitesides' group could reproduce our results and also found a factor of 600 in the charge transport rates when X = H was replaced with X = Br when small junctions were used[47].

Here, we show that the value of $\beta$ of molecular wires with an alkyl chain backbone can be reduced from 0.75 to 0.25 Å$^{-1}$, in effect turning them from insulators into conductors without changing the chemical structure of the backbone of the molecular wire, by introducing one distal polarizable atom at one end of the

molecular wire of the form $HS(CH_2)_nX$. Changing X from H to I in the long $S(CH_2)_{18}X$ molecular wire gives a factor of $10^{\approx 5}$ increase in $J$. For $S(CH_2)_{10}X$, the currents change by a factor of $10^{\approx 2}$. As we will discuss below, these observations cannot be explained by changes in the molecule–electrode interfaces, or contact resistances, alone. While we have shown before that the halide group affects the current and $\varepsilon_r$ in $Ag-S(CH_2)_{11}X//GaO_x/$EGaIn junctions[45], here we demonstrate that the value of $\beta$ can be controlled by changing X without the need to modify the chemical structure of the molecular backbone. This change in $\beta$ explains why the largest change in current is found for the longest molecules studied in this work. On the basis of experimental and theoretical data, we discuss how introducing this polarizable atom changes the electrostatic potential profile of the tunnelling barrier, the $\varepsilon_r$ of the junction, and the contact resistance (or $\Gamma$), which are important to consider when modifying tunnelling efficiency across molecular wires.

## Results

**The junctions.** Figure 1 shows a schematic illustration of the Ag–S$(CH_2)_nX//GaO_x$/EGaIn junctions, and indicates how the coupling and energy level alignment (i.e., $\Gamma$ and $\delta E_{ME}$) change with X as discussed in detail below. The schematic also includes the equivalent circuit consisting of the contact resistance ($R_C$, in m$\Omega$ cm$^2$) in series with a parallel combination of the SAM resistance ($R_{SAM}$, in $\Omega$ cm$^2$) and the capacitance of the SAM ($C_{SAM}$, in $\mu$F/cm$^2$) in the junction. The equivalent circuit and the associated physical meaning of each circuit component has been explicitly discussed in our previous work[41] (and is summarized in Supplementary Section 7). Briefly, the $R_C$ includes the resistances of the contacts of the SAM with the top and bottom electrodes, and the resistance of the electrodes and wires connecting the junction with the electrometers. The SAM itself behaves as a capacitor ($C_{SAM}$) with associated resistance ($R_{SAM}$) as expressed in Eqs. (2) and (3). It highlights that the junctions are essentially parallel plate capacitors in which dielectric behaviour depends on the chemical structure of the junctions, which, as we show below, is also important to explain tunnelling rates. All SAM precursors were synthesized following previously reported methods and characterized with $^1$H NMR, $^{13}$C NMR, and mass spectroscopy (Supplementary Sections 1 and 2). The SAMs were formed on template-stripped Ag electrodes using well-established methods

and the junctions were completed with cone-shaped GaO$_x$/EGaIn top contacts[53] (Supplementary Sections 3 and 6). Previously, we have reported that for EGaIn junctions with $S(CH_2)_nX$ SAMs (there only $n = 11$ was studied) the measured current increased by three orders of magnitude and the value of $\varepsilon_r$ increased by a factor of 4, when X was changed along the halogen series from F to I[45]. However, the evolution of $J$ and $\varepsilon_r$ with increasing molecular length and the corresponding $\beta$ values for different X have so far not been studied. Here we address whether this increase in current is caused by changes in $R_C$ or by changes from coherent tunnelling to incoherent process (i.e., $\beta$). Changing the value of $n$ for each X allows us to investigate in detail how and why $\beta$ changes as a function of X while keeping the nature of the molecule–electrode interfaces and the molecular backbone the same.

**Characterization of the SAMs.** We characterized the SAMs on Ag with $n = 14$ for X = H, F, Cl, Br, or I, and $n = 10, 14,$ or 18 for X = Br with angle-resolved X-ray photoelectron spectroscopy (ARXPS) and molecular dynamics (MD) simulations (for all combinations of $n$ and X) and all results are summarized in Table 1 (see Supplementary Sections 4 and 5 for details). Figure 2a shows a representative snapshot from the MD simulations of Ag–S$(CH_2)_{14}$I SAM with computed molecule heights in excellent agreement with film thicknesses $d_{SAM}$ measured by XPS (Fig. 2b, c and Table 1) indicating that the $S(CH_2)_{14}$I precursor readily forms dense layers with all molecules in a fully-upright position. We determined the relative values of surface coverage ($\Psi_{SAM}$) with XPS, which confirms that all SAMs have indistinguishable packing densities (Fig. 2b, c) within experimental error. The value of $d_{SAM,MD}$ increases by about 1.4 Å overall on increasing van der Waals radius of X from H to I (Supplementary Table 1), but this small increase falls within the experimental error of $d_{SAM,XPS}$ (Fig. 2b). Figure 2c shows that for X = Br, $d_{SAM,XPS}$ increases linearly with $n$ with a slope of $1.5 \pm 0.1$ Å per carbon (solid blue line, error represents standard error from linear fit), which is in close agreement with the MD value of $1.3 \pm 0.1$ Å per carbon (dashed blue line; see Supplementary Fig. 11 for $d_{SAM,MD}$ values of all the SAMs). Figure 2d shows the packing energies per molecule ($E_{mol,MD}$, in eV) and per methylene CH$_2$ unit ($E_{meth,MD}$, in meV) extracted from the MD calculations. The values of $E_{mol,MD}$ and $E_{meth,MD}$ improve slightly as X shifts from H ($-1.8 \pm 0.1$ eV per molecule) to Br ($-2.4 \pm 0.2$ eV per molecule), which is due to the increasing intermolecular van der Waals interaction. For SAMs with X = I, the

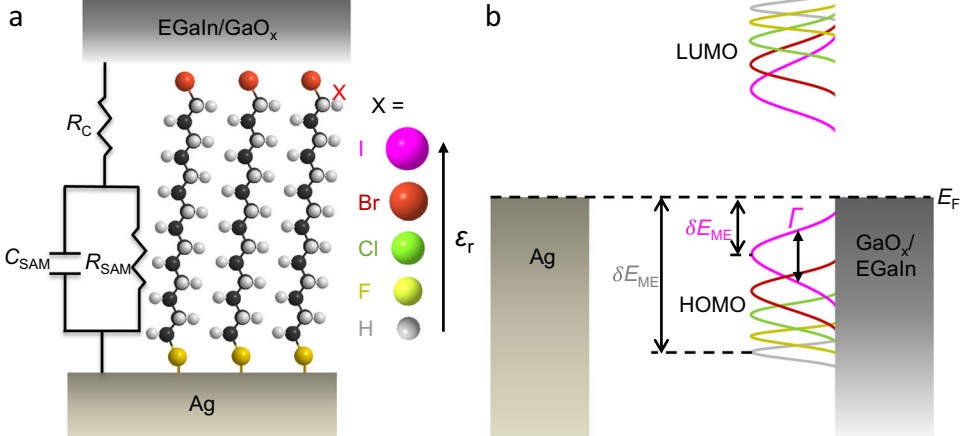

**Fig. 1 The junctions, equivalent circuit, and energy level diagram. a** Schematic illustration of the Ag–S$(CH_2)_nX//GaO_x$/EGaIn junction (shown for $n = 14$, EGaIn is for eutectic alloy of Gallium and Indium, "-" represents covalent bond, "//" represents non-covalent contact, "/" means the interface between GaO$_x$ and EGaIn) together with the equivalent circuit diagram. In this work we investigated junctions with $n = 10, 12, 14, 16,$ or 18, and X = H, F, Cl, Br, or I. **b** Energy level diagram of the junction showing how the coupling strength between molecules and electrodes ($\Gamma$) and tunnelling barrier height ($\delta E_{ME}$) change with X.

**Table 1 Summary of properties of the Ag–S(CH₂)ₙX SAMs.**

| X and $n$ | $\Psi_{SAM,XPS}$ (nmol/cm²)[a] | $d_{SAM,XPS}$ (Å) | $d_{SAM,MD}$ (Å) | $E_{mol,MD}$ (eV) | $\Phi_{SECO}$ (eV)[b] | $\Phi_{DFT}$ (eV) | $\varepsilon_r$ | $\varepsilon_{DFT-VdW}$ |
|---|---|---|---|---|---|---|---|---|
| $n = 14$, X = H | 0.74 | 18 | 20.4 ± 0.5 | −1.8 ± 0.1 | 3.98 | 3.47 | 2.9 ± 0.3 | 2.2 |
| $n = 14$, X = F | 1.0 | 21 | 21.1 ± 0.3 | −2.0 ± 0.1 | 4.43 | 5.44 | 2.5 ± 0.6 | 2.1 |
| $n = 14$, X = Cl | 0.86 | 21 | 21.5 ± 0.3 | −2.2 ± 0.2 | 5.02 | 5.25 | 3.0 ± 0.2 | 2.2 |
| $n = 14$, X = Br | 1.1 | 20 | 21.7 ± 0.3 | −2.4 ± 0.2 | 4.77 | 5.18 | 4.7 ± 0.9 | 2.3 |
| $n = 14$, X = I | 1.2 | 21 | 21.8 ± 0.3 | −2.2 ± 0.1 | 4.68 | 4.96 | 8.9 ± 1.6 | 2.4 |
| $n = 10$, X = Br | 1.0 | 15 | 16.2 ± 0.4 | −1.7 ± 0.1 | 4.62 | – | 4.4 ± 0.4 | – |
| $n = 18$, X = Br | 1.1 | 29 | 26.8 ± 0.3 | −3.0 ± 0.2 | 4.66 | – | 4.6 ± 0.2 | – |

[a]The $\Psi_{SAM,XPS}$ are relative to $\Psi_{SAM}$ of Ag–S(CH₂)₁₄F SAM as measured by XPS.
[b]The experimental error is ±0.05 eV.

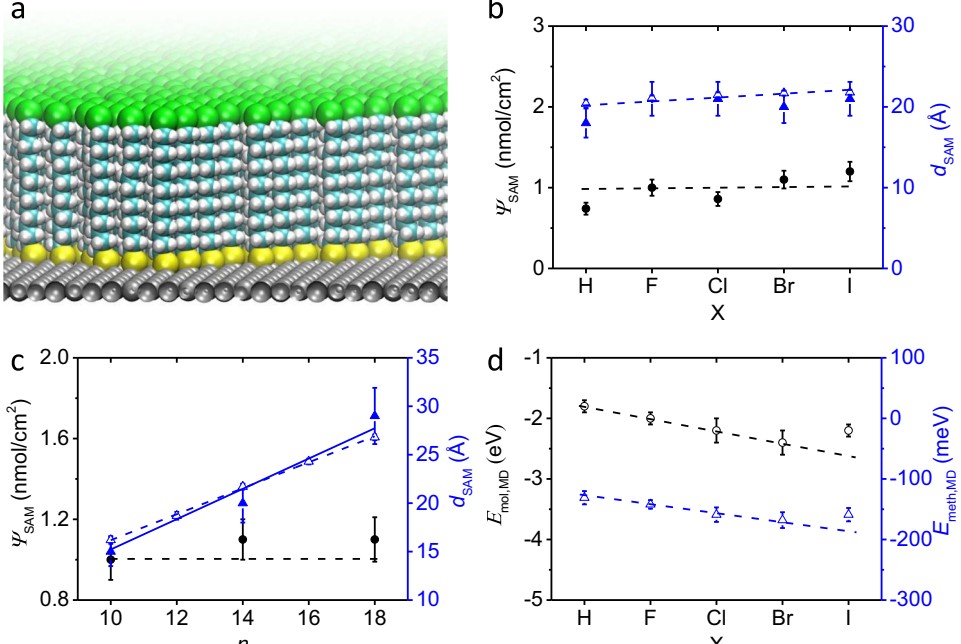

**Fig. 2 Characterization of the self-assembled monolayers (SAMs). a** Representative slice-through of a large-area Ag–S(CH₂)₁₄I SAM structure calculated by molecular dynamics (MD) computer simulations. **b** Surface coverage ($\Psi_{SAM}$) of Ag–S(CH₂)₁₄X SAMs as a function of X determined with angle-resolved X-ray photoelectron spectroscopy (ARXPS, filled circles) and thickness of SAM ($d_{SAM}$) determined with ARXPS (filled triangles) and MD (empty triangles). **c** $\Psi_{SAM}$ of Ag–S(CH₂)ₙBr SAMs as a function of $n$ determined with ARXPS (filled circles) and $d_{SAM}$ determined with ARXPS (filled triangles) and MD (open triangles). The solid and dashed blue lines are linear fits to the experimental and MD data with $R^2$ of 0.94 and 0.99, respectively. The horizontal dashed line in panels **b** and **c** indicates the $\Psi_{SAM}$ used in the MD calculations. **d** Computed MD packing energy per molecule $E_{mol,MD}$ and per methylene –CH₂– unit $E_{meth,MD}$ of Ag–S(CH₂)₁₄X SAMs as a function of X. Dashed lines are guides to the eye. The errors on the XPS data represent instrumental and fitting errors of 10% in total (see Section S4). The error bars in the MD data represent the standard deviations in the time- and molecule-averages calculated across 500 snapshots taken during the final 50 ns of 100 ns of room temperature MD of 128-molecule Ag–S(CH₂)₁₄X SAMs with the average experimental coverage of 1 nmol/cm² on Ag(111).

packing energies weaken slightly due to small competing effects caused by mild steric repulsion between the large I headgroups. These observations confirm that the halogen functionality does not significantly disrupt the supramolecular structure of the SAM. Finally, we determined the energy level alignment of the SAMs on Ag using ultra-violet photoemission spectroscopy in Supplementary Section 4 and Supplementary Fig. 10, which we used to validate our density-functional theory (DFT) calculations as discussed in more detail below (Table 1).

**Electrical characterization of the junctions.** To study how the halogen functionality affects the tunnelling rates across the SAMs, we measured the electrical characteristics of the junctions as a

function of X and $n$ using $J(V)$ measurements and impedance spectroscopy. The SAMs were contacted with cone-shaped GaOₓ/EGaIn electrodes following a previously reported method[53]. To minimize leakage currents and to ensure that molecular effects dominate the junction characteristics, we used junctions with a small contact area of ~350 μm², as large junctions suffer from leakage currents across defective sites[52]. We recorded statistically large numbers of $J(V)$ curves to determine the Gaussian log-average $J(V)$ curves, $<\log_{10}|J|>_G$, and associated Gaussian log-standard deviations ($\sigma_{log,G}$) which are plotted in Fig. 3a for junctions with X = F, and in Fig. 3b for junctions with X = I, for $n = 10$–18 (all Gaussian log-average $J(V)$ curves and histograms of the $\log_{10}|J|$ at ±0.5 V are given in Supplementary Section 6). Clearly, the tunnelling rates are more attenuated for X = F than for X = I. Figure 3c

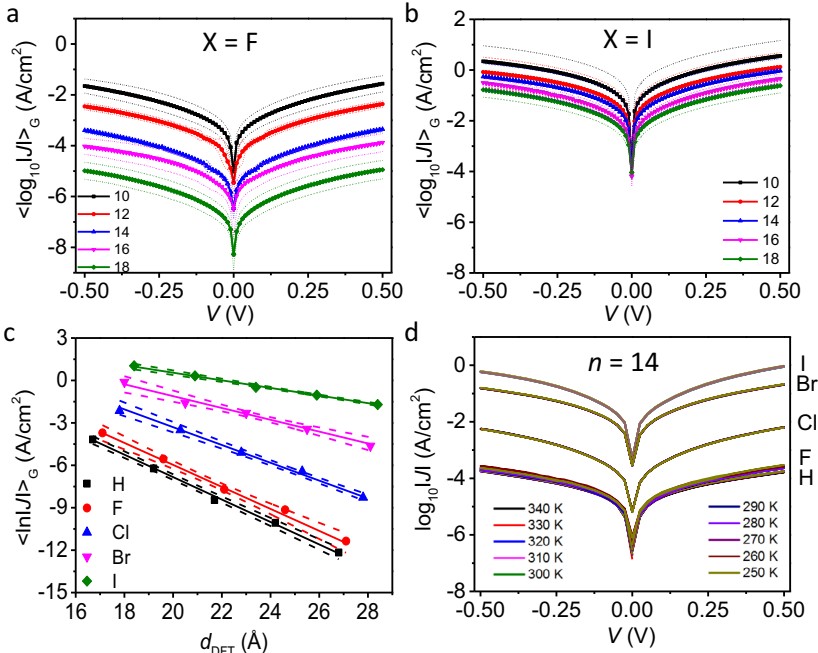

**Fig. 3 Electrical characterization of junctions.** Gaussian log-average values of the current densities $<\log_{10}|J|>_G$ vs. applied bias $V$ obtained from Ag–S$(CH_2)_nX$//GaO$_x$/EGaIn junctions with X = F (**a**) or I (**b**) and $n$ = 10 (solid black line), 12 (solid red line), 14 (solid blue line), 16 (solid pink line), and 18 (solid green line). The dashed-line error bars represent the Gaussian log-standard deviation, $\sigma_{\log,G}$. **c** Decay plots of $<\log_{10}|J|>_G$ at –0.5 V against $d_{SAM,MD}$ with X = H (black square), F (red circle), Cl (blue triangle), Br (pink inverted triangle), or I (green diamond). The solid lines are fits to Eq. (1). The dashed lines represent the 95% confidence bands. **d** Plots of $\log_{10}|J|$ vs. $V$ as a function of $T$ ($T$ = 250–340 K) recorded from Ag–S$(CH_2)_{14}$X//GaO$_x$/EGaIn junctions.

shows the decay of $<\log_{10}|J|>_G$ at –0.5 V as a function of $d_{SAM,MD}$ (Supplementary Section 5 and Supplementary Table 1) for all X. The solid lines are fits to Eq. (1) from which we determined the values of $\beta$ which are listed in Supplementary Table 7. Supplementary Figure 18 shows the plot of $\beta$ vs. X. Interestingly, the value of $\beta$ steadily decreases from $0.75 \pm 0.01$ Å$^{-1}$ for X = H—a typical value for tunnelling along alkyl chains—to $0.25 \pm 0.01$ Å$^{-1}$ for X = I which is a typical value for tunnelling along $\pi$-conjugated molecules (the error in $\beta$ represents the standard error of the fit to Eq. 1). We measured the $J(V)$ characteristics as a function of temperature, $T$ in K, of Ag–S$(CH_2)_{14}$X//GaO$_x$/EGaIn junctions for all X using top electrode of EGaIn confined in a microfluidic network in poly-dimethylsiloxane following a previous reported method[41] (see Supplementary Section 6 for details). Figure 3d shows that the tunnelling rates are independent of $T$ in the range of $T$ from 250 to 340 K, which is consistent with coherent off-resonant tunnelling[54].

**Dielectric constant of the junctions.** To characterize the dielectric response of the junctions, we conducted impedance spectroscopy using a sinusoidal voltage perturbation with an amplitude of 30 mV around 0 V in the frequency range of 100 Hz to 1.00 MHz and the data were fitted to the equivalent circuit shown in Fig. 1a following a previously reported method[41] (see Supplementary Section 7 for details). Supplementary Figure 23 shows the Bode, Nyquist, and the corresponding phase angle ($\varsigma$) vs. frequency ($f$) plots along with the fits to the equivalent circuit (Supplementary Tables 8–10 list all fitting results). Figure 4a shows that $R_C$ decreases by a factor of 5 when X is changed from H or F to I while $R_C$ is independent of $n$ (Fig. 4b). This change in $R_C$ indicates that $\Gamma$ substantially increases as a function of X. This increase in $\Gamma$ can be rationalized by the increase in polarizability $\alpha$ and associated induced dipoles as a function of X resulting in an increase in the van der Waals interaction strength between the SAM and the top contact[45]. Supplementary Table 9 shows the

decrease of $R_{SAM}$ with X which is mainly caused by lowering of $\delta E_{ME}$ and increase of $\Gamma$ (see the "DFT Calculations" section). Frisbie and co-workers[9,38] have shown that a decrease in $R_C$ by increasing the work function of the bare metal electrodes increases the conductivity of molecular junctions (with H or S terminal atoms), which was mainly driven by a large increase in $\Gamma$, with changes in $\delta E_{ME}$ and $\beta$ playing only a minor role.

To confirm the consistency between the $J(V)$ and impedance measurements, we determined the value of $\beta$ from the impedance measurements for junctions with X = Br. The value of $R_{SAM}$ increases exponentially with $n$ (Eq. 2)

$$R_{SAM} = R_{SAM,0}(V)e^{\beta d_{SAM,MD}} = R_{SAM,0}(V)10^{\beta d_{SAM,MD}/2.303} \quad (2)$$

where $R_{SAM,0}$ is a pre-exponential factor. Figure 4c shows the plot of $\log_{10}R_{SAM}$ vs. $n$ along with a fit to Eq. (2) from which we extracted the values of $\beta = 0.41 \pm 0.03$ Å$^{-1}$ and $\log_{10}R_{SAM,0} = -2.0 \pm 0.2$ Ω/cm$^2$ (or $R_{SAM,0} = 1.0 \times 10^{-2}$ Ω/cm$^2$). The value of $R_{SAM,0}$ is essentially equivalent to $J_0$ (defined in Eq. 1) derived from a current decay plot at 30 mV (since the sinusoidal perturbation used in the impedance measurements was 30 mV). The value of $J_0$ at 30 mV is $5.1 \pm 2.0$ A/cm$^2$ and the $\beta = 0.46 \pm 0.03$ Å$^{-1}$ (Supplementary Fig. 17). $R_{SAM,0} \approx V/J_0 = 0.59 \times 10^{-2}$ Ω/cm$^2$, which is within a factor of 2 of the value measured with impedance spectroscopy ($R_{SAM,0} = 1.0 \times 10^{-2}$ Ω/cm$^2$). The contribution of $R_C$ is minor since $R_C$ is a parallel circuit element, but it is included in $J_0$. The $\beta$ and $R_{SAM,0}$ values are, within error, the same as the values determined with the $J(V)$ measurements.

To gain further insight into the dielectric properties of the junctions, we used the parallel plate capacitor equation (Eq. 3) to determine $\varepsilon_r$ as a function of X and $n$

$$C_{SAM} = \varepsilon_0\varepsilon_r \frac{A_{geo}}{d_{SAM,MD}} \quad (3)$$

wherein $\varepsilon_0$ is the vacuum permittivity and $A_{geo}$ is the geometrical

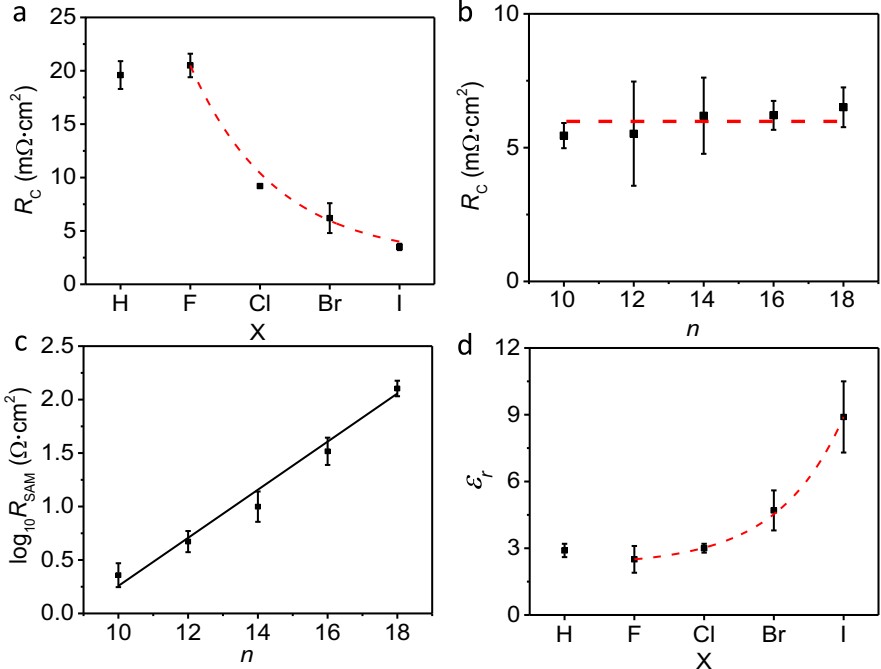

**Fig. 4 Characterization of junctions with impedance spectroscopy. a** Contact resistance $R_C$ vs. X for Ag-S(CH$_2$)$_{14}$X//GaO$_x$/EGaIn junctions at DC of 0 V and sinusoidal perturbation of 30 mV. Log-resistance of SAM, log$_{10}R_{SAM}$, (**b**) and $R_C$ (**c**) vs. $n$ for Ag-S(CH$_2$)$_n$Br//GaO$_x$/EGaIn junctions. The solid black line represents a fit to Eq. (2). **d** Corresponding dielectric constant $\varepsilon_r$ vs. X for Ag-S(CH$_2$)$_{14}$X//GaO$_x$/EGaIn junctions. The error bars are the standard deviations of three independent measurements. Dashed lines are visual guides.

area of the junction. Figure 4d shows that $\varepsilon_r$ increases by a factor of 3 when changing X from H or F to I, yet $\varepsilon_r$ is independent of $n$ (Supplementary Tables 9–10). Although this factor 3 increase in $\varepsilon_r$ is expected for bulk systems which can be described via the Clausius–Mosotti relation[55], this observation cannot be explained as an intrinsic electrostatic property of the molecular wires, as we will show and discuss below. Moreover, even though we can quantify the contributions of different circuit components from impedance spectroscopy, how these components are influenced by each other are not directly revealed. Therefore, we referred to DFT and Landauer modelling for further explanations.

**DFT calculations**. To provide further microscopic insight into the electrostatic properties and electronic structure of the molecular wires, we performed first-principles calculations based on DFT using the VASP code[56] and a $3 \times 2\sqrt{3}$ Ag surface unit cell containing four molecules arranged in a herringbone pattern (see Supplementary Section 8 for full details).

Figure 5a shows that the shape of the potential energy towards the tail of the alkyl chain strongly depends on the functionalization at the X-site. Specifically, the vacuum level changes with X functionalization owing to the polarity of the C–X bond, which translates into a change of the Ag work function, $\Phi$, as expected for SAMs with different tail groups[45,46,57,58]. Comparing the DFT-calculated $\Phi$ to the experimental $\Phi$, it can be seen that the agreement is good for all terminations except X = F (Fig. 5b). We tentatively ascribe the quantitative deviations to the often observed overestimation of polar effects in periodic DFT calculations of metal–SAM interfaces due to the assumption of perfect molecular order and periodicity[59], while practical systems have defects (e.g., step edged, grain boundaries, or phase domains) and are dynamic in nature[52,60].

Figure 5c reports the density of states (DOS) projected onto the molecular part of Ag-S(CH$_2$)$_{14}$X. All systems show a feature at ~1.5 eV (marked by * in Fig. 5c) that is due to Ag–S hybridization

(Supplementary Section 8). Interestingly, we find that lower lying occupied states as well as the lowest unoccupied state strongly shift in energy (on the order of 1–2 eV) with varying X (see arrows in Fig. 5c). These energy shifts clearly correlate with X functionalization and increase in magnitude along the halogen series, so that the X = I SAM shows pronounced new features close to the band edges when compared to the X = H or X = F SAM. Figure 5d shows the DOS projected onto just the X-site in Ag-S(CH$_2$)$_{14}$X, which confirms that these new occupied and unoccupied states are due to the halogen functionalization. For both the occupied and unoccupied parts of the DOS, these halogen-derived states do partially overlap in energy with other features but are localized primarily at the tail of the SAM (Supplementary Section 8). The X groups are not redox-active, and even for X = I the HOMO is still ~1.7 eV below $E_F$ (Fig. 5c, d). Hence, the HOMO cannot enter the applied bias window of ±0.5 V (note the molecules with $n = 10$ tend to break down at higher voltages)[61].

Finally, we determined $\varepsilon_r$ as a function of X for the free-standing and hydrogen-terminated HS(CH$_2$)$_{14}$X SAMs using previously reported protocols[44,62] (see Table 1 and Supplementary Section 8). Hereby, $\varepsilon_r$ is calculated from the change in the dipole moment induced by an applied static electric field, with the atomic positions fixed at their equilibrium positions. Thus, the $\varepsilon_r$ obtained in such a manner represents the instantaneous response of the electronic charge density to a static electric field. In contrast to the above-discussed experimental results, we find that $\varepsilon_r$ hardly changes with X functionalization in our DFT calculations. This result is expected from purely electrostatic reasoning and fully in line with previous work by various groups[42–44]. Briefly, in these studies it has been shown from electrostatic and DFT calculations that varying the molecular polarizability of the SAM-forming molecules does not result in significant changes of $\varepsilon_r$ in the densely packed conjugated SAMs due to depolarization effects arising from the neighbouring molecular dipoles in the SAM[42–44]. Therefore, the calculations show that tuning the molecular

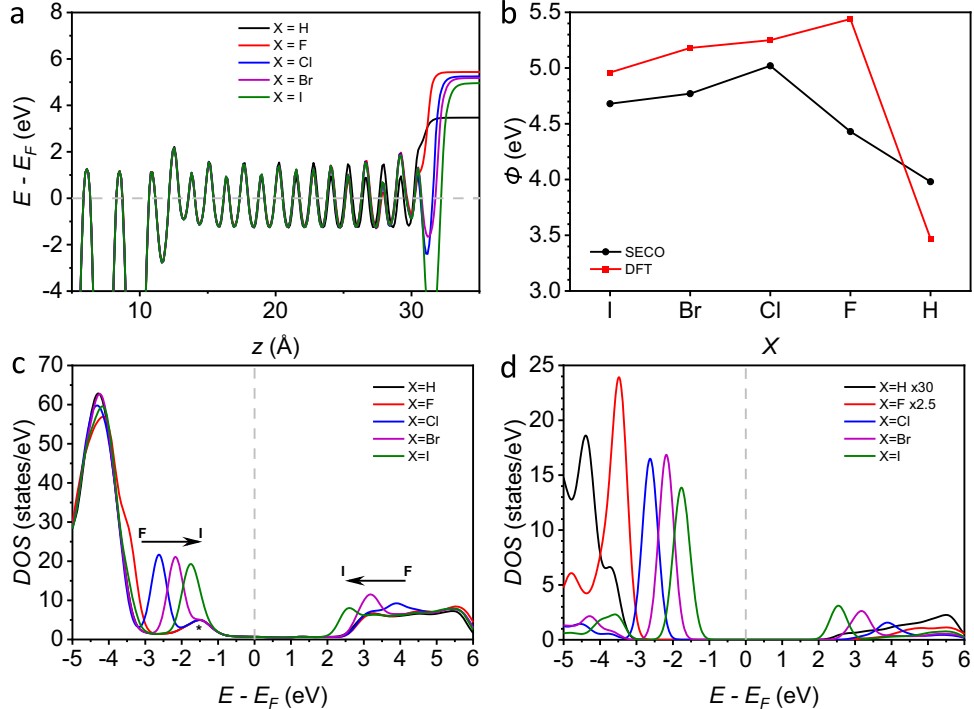

**Fig. 5 Density-functional theory (DFT) calculations. a** DFT-calculated plane-averaged electrostatic potential of Ag(111)–S(CH$_2$)$_{14}$X where X = H, F, Cl, Br, or I, along the surface-normal coordinate. **b** Work function ($\Phi$) of the SAMs, calculated from DFT (red squares) and measured experimentally (black dots). Density of states (DOS) projected onto the molecular backbone (**c**) and onto the X-site (**d**) of Ag(111)–S(CH$_2$)$_{14}$X. Note that X contributes very little to the band edges for X = H, F.

polarizability by changing X does not strongly impact the calculated $\varepsilon_r$ of the HS(CH$_2$)$_{14}$X SAMs.

As pointed out by Natan et al.[43], a competition between suppression of in-plane polarization and enhancement of out-of-plane polarization occurs in SAMs. The suppression dominates for densely-packed SAMs and, thus, the substituent X should not affect the calculated $\varepsilon_r$ of the SAM in sharp contrast to our herin reported experimental findings. Our calculations as well as previous theoretical studies, however, only probe the intrinsic dielectric properties of the isolated highly organized SAM without contacts. The interaction between the SAM and the top electrode that is naturally present in the experimental determination of $\varepsilon_r$ could affect the dielectric behaviour of the junction considerably, which would be consistent with the experimentally recorded trends for the $R_C$ shown in Fig. 4a. Given the high electric fields on the order of GV/m and the polarizable nature of X, the substituents may be partially charged during charge transport (especially iodines are well-known to readily accommodate electrons)[63,64]. We note that previously reported DFT calculations of $\varepsilon_r$ of the HS(CH$_2$)$_{11}$X SAMs[45] were incorrect due to simulation artefacts of uncompensated dipoles in the unit cell, which created a spurious correlation with experimentally measured $\varepsilon_r$ values.

**Single-level Landauer model.** In the following, we discuss our results in the context of commonly used models to interpret charge transport through the S(CH$_2$)$_n$X molecular junctions. The single-level Landauer model is frequently used to model the current flowing across molecular tunnel junctions[65]. Here we modelled the current using the following expression:

$$I = \frac{Nq}{h} \int \int_{-\infty}^{\infty} dEdE' D_{E'}(E) G_{\delta E_{ME}}(E') \frac{\gamma_L \gamma_R}{\gamma_L + \gamma_R} [f_L(E) - f_R(E)] \quad (4)$$

where $\gamma_L$ and $\gamma_R$ are the tunnelling rates between the molecule and the left and right electrodes (respectively), $D_{E'}(E)$ is the electronic density of states of the molecular level having the shape of Lorentzian and is given by

$$D_{E'}(E) = \frac{\frac{\gamma}{2\pi}}{\left(E - \left(E' + \left(\eta - \frac{1}{2}\right) \times V\right)\right)^2 + \left(\frac{\gamma}{2}\right)^2} \quad (5)$$

centred at energy $E' + \left(\eta - \frac{1}{2}\right)V$, where $\eta = V_R/(V_L + V_R)$ is the voltage division parameter accounting for the capacitive coupling with the left and right electrodes, and with a level width $\gamma = \gamma_L + \gamma_R$. The $f_L(E)$ and $f_R(E)$ are the Fermi functions representing the electronic occupation of the left and right electrodes, respectively, which are given by[65]

$$f_{L,R}(E) = \frac{1}{1 + \exp\left[\frac{E \pm \frac{V}{2}}{K_B T}\right]} \quad (6)$$

Equations 4–6 provide the model to which we fitted the experimental data. In addition, we attached a Gaussian to the model with the inherent dispersion ($\sigma$) of the molecular level energy ($\delta E_{ME}$) in an ensemble of molecules (rather than a single-molecule junction), as given by the following expression:

$$G_{\delta E_{ME}}(E') = A \exp\left(\frac{\left(E' - \delta E_{ME}\right)^2}{2\sigma^2}\right) \quad (7)$$

We accounted for the behaviour of a group of molecules by setting the number of such molecules fixed at $N = 150$. As obviously seen in the above model, the current is directly dependent on $\gamma_L \times \gamma_R$. There is a trade-off between the Gaussian and the density of states which is in the shape of a Lorentzian centred at the energy level $\delta E_{ME}$. All the molecules appeared to be symmetric and what accounts for the difference in conductance is the terminal atom on the molecular unit. In this case, the ligands were X = H, F, Cl, Br, or I. Therefore five different set

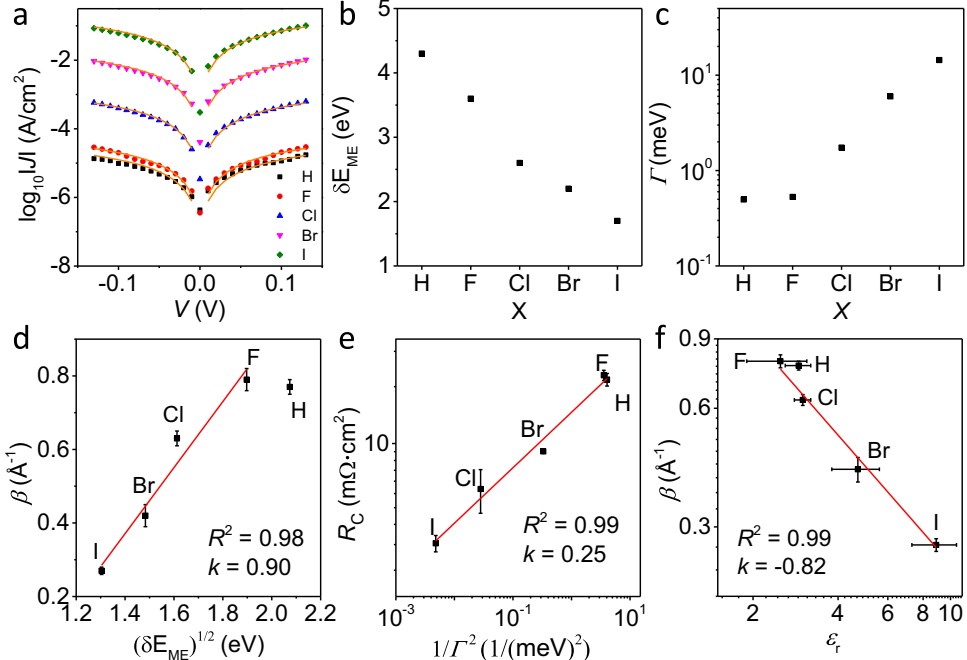

**Fig. 6 Single-level Landauer model analysis. a** The modelled current through the Ag–S(CH$_2$)$_{14}$X//GaO$_x$/EGaIn junctions using Landauer theory (orange solid lines are Landauer fits, symbols represent experimental data). The values of tunnelling barrier height ($\delta E_{ME}$) (**b**) and the coupling strength ($\Gamma$) (**c**) used for modelling the current through the junctions. **d** Tunnelling decay coefficient $\beta$ vs. $\sqrt{\delta E_{ME}}$ with a linear fit (red line), the error bars represent the standard deviations of the $\beta$ values from linear fits to Eq. (1). **e** Double-log plot of $R_C$ vs. $1/\Gamma^2$ ($R_C$ represents contact resistance) where the red line is a power-law fit with a slope of 0.25 and $R^2 = 0.99$, error bars of $R_C$ represent the standard deviations of three independent measurements. **f** Double-log plot of $\beta$ vs. $\varepsilon_r$ where the red line is a fit with a slope of $-0.82$ and $R^2 = 0.99$. The error bars of $\beta$ represents the same as panel **b**, and of $\varepsilon_r$ represent the standard deviations of three independent measurements.

of fittings were done for each X, fixing $\delta E_{ME}$ to the values extracted from DFT and leaving $\gamma_L$, $\gamma_R$, $\eta$ and $\sigma$ as fitting parameters to obtain best fits to the data of junctions of Ag–S(CH$_2$)$_{14}$X//GaO$_x$/EGaIn (Supplementary Table 12). Figure 6a shows the fits of the theoretical model (orange lines) to the experimental data (symbols) for each S(CH$_2$)$_{14}$X molecule. Figure 6b, c shows the two parameters that vary across molecules: the energy $\delta E_{ME}$ of the frontier orbital (extracted from DFT), which decreases from 4.3 to 1.7 eV on moving through the sequence H–F–Cl–Br–I (Fig. 6b), and the overall tunnelling rate through the junction (i.e., the $\Gamma$), defined as $\Gamma = \frac{\gamma_L \gamma_R}{\gamma_L + \gamma_R}$, which increases exponentially along the halogen sequence (Fig. 6c) and accounts for the observed exponential increase of the current through the junctions.

Figure 6d shows a linear relationship between calculated $\sqrt{\delta E_{ME}}$ and measured $\beta$, which agrees with commonly used coherent tunnelling models[2,5,18] including the Simmons model which also accounts for $\varepsilon_r$. However, the Simmons model also predicts a decrease of the tunnelling rates with increasing $\varepsilon_r$ due to a reduction of the image charge effects in the electrodes due to screening within the SAM[18,66]. This reduction of image charge in effect increases $\delta E_{ME}$ and, consequently, $\beta$, but we observe the opposite trend. Using the same model, a reduction of the effective electron mass could also account for an increase in tunnelling rates, but it is not clear how the effective electron mass would change as a function of X with the essentially localized features derived from the HOMO and LUMO. Furthermore, Vilan[67] argued that changes in the electron mass are equivalent to changes in the $\delta E_{ME}$ within the Simmons model, which further complicates the interpretation of our findings within this framework.

The experimentally determined values of $R_C$ have been related to $\Gamma$ as $R_C \propto \Gamma^{-2}$ (ref. [30]), i.e., the coupling of the molecules with the

electrode we have determined above (Fig. 6e). To test whether this holds for the SAMs studied here, Fig. 6e shows a double-log plot of $R_C$ vs. $1/\Gamma^2$, indicating that our results can be explained, at least qualitatively, using this picture: changes in both $\delta E_{ME}$ and $\Gamma$ can lower $\beta$, in accordance with findings by others[27–35]. This approach, however, does not capture the observed changes in the dielectric response of the junctions directly, and, of course, it does not explicitly account for the local changes in the electrostatic potential profile induced by X observed in the DFT calculations; these effects are essentially compensated by the large change in $\Gamma$ of 29 times.

An interesting finding was reported by Berlin and Ratner[68] based on an alternative model to describe tunnelling across barriers with charge traps, with the finding that $\beta \propto 1/\sqrt{\varepsilon_r}$. In this framework, the distance dependence of the conductance is related to a thickness-dependent barrier akin to the one inherent to the Simmons model[67,69]. In their model, however, the barrier arises from the presence of localized charge traps along the path of charge migration leading to a non-linear potential drop between the macroscopic leads. Figure 6f shows the linear relation of the double-log plot of $\beta$ vs. $\varepsilon_r$ with a slope of $-0.82$ which is lower than the expected $-0.5$ from the model by Berlin et al.[68], but note that a change in the contact resistance or further changes in the barrier shape are not taken into consideration in this model. In our experiments, however, the contact resistance changes and our DFT calculations show that the barrier shape at the SAM//top electrode interface is affected by X.

## Discussion

This work shows that substitution of a single highly polarizable atom can have a pronounced effect on the energy level alignment, charge transport rate, and dielectric response of molecular junctions. We were able to tune $\beta$ over a wide range from 0.25 to 0.75 Å$^{-1}$ across

saturated alkyl chains by changing one atom per molecule inside large-area ($\sim$350 $\mu m^2$)[52] junctions. The largest effects of X on $J$ are found in the longest molecules of $S(CH_2)_{18}X$ where the $J$ increases by a factor of $10^{\approx 5}$ when X changes from H to I. In contrast, for the shortest molecule $S(CH_2)_{10}X$, $J$ increases by a factor of $10^{\approx 2}$, indicating that the observed changes in $J$, and the corresponding values of $\beta$, are driven by more than just changes in the interfaces, which has been discussed previously by Frisbie and co-workers[9,38]. Combining experiment with DFT and Landauer charge transport models, we established three factors that contribute to the dramatic change in $\beta$ of these aliphatic halogenated junctions with varying X and associated increase in $\alpha$ and $\varepsilon_r$: (1) The HOMO-LUMO gap[5,45] and associated $\delta E_{ME}$ is reduced which lowers $\beta$, (2) the shape of the tunnelling barrier is modified at the SAM-top electrode interface, resulting in larger potential drops at this interface, and (3) the electronic coupling $\Gamma$ of the molecular orbitals with the electrodes increases (potentially because of an increase in the van der Waals interactions along the halogen series).

In a broad context of widely used charge tunnelling mechanisms, our findings point out their limitations highlighting the need for improved models that take dielectric (or collective) effects of the junctions into consideration. Specifically, the popular Simmons model predicts that image charge effects in the electrodes are reduced with increasing $\varepsilon_r$, resulting in lowering of the tunnelling rates[67], which is in sharp contrast to what we find. Superexchange models[19–21] also fail to explain our observations, since the coupling between the molecular repeat unit (i.e., the $CH_2$ units) was not changed here. Conversely, the Landauer model[65] could explain our results at least qualitatively, but not quantitatively. This is because it does not treat electrostatic effects in the junctions explicitly and self-consistently which resulted in our case in a large increase in the values of $\Gamma$ (29 times) even though the $R_C$ only changed by a factor of five in our experiments. Interestingly, a mechanism proposed by Berlin and Ratner[68] that is based on charge traps provides a hypothesis for how the value of $\beta$ could decrease with increasing $\varepsilon_r$. Although the physical interpretation differs as the tunnelling behaviour is explained in terms of charge traps rather than the electrostatic response of the SAM inside the junction, this line of thought stimulates further theoretical and experimental testing of the presence of "impurities"—here in the form of polarizable atoms—as charge carriers that move across the energy band profiles. Although our findings suggest a correlation between $\beta$ and $\varepsilon_r$, the increase of $\varepsilon_r$ as a function of X could not be reproduced in our DFT calculations, which may be because the calculations do not take the SAM–metal interface into consideration, and perhaps other factors are important such as (partial) charging of highly polarizable molecules during charge transport inside the junctions. To summarize, our work proposes an effective way of tuning the tunnelling rates across molecular junctions without chemically altering the backbone of the molecules and highlights the importance of understanding dielectric effects in these junctions. We hope that our findings will stimulate further experimental and theoretical investigations towards establishing improved transport mechanisms for junctions in their in situ physicochemical environment and electronic states inside working devices.

## Data availability
The data that support the findings of this study are available within the article and Supplementary Information file, or at https://doi.org/10.7910/DVN/WZAIGU. Source data are provided with this paper.

## Code availability
The code for the analysis of $J(V)$ results in this study is available at https://doi.org/10.7910/DVN/WZAIGU.

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

## Acknowledgements

We acknowledge fruitful discussions with Ayelet Vilan (Weizmann Institute of Science) and Gemma Solomon (University of Copenhagen). Fundings by the Ministry of Education (MOE) for supporting this research under award No. MOE2019-T2-1-137 and R-143-000-B30-112 are acknowledged. Prime Minister's Office, Singapore under its Medium sized centre programme is also acknowledged for supporting this research. The authors would furthermore like to acknowledge the Singapore Synchrotron Light Source (SSLS) for providing the facilities at the Surface, Interface and Nanostructure Science (SINS) beam line under NUS core support C-380-003-003-001. The Laboratory is a National Research Infrastructure under the National Research Foundation Singapore. We moreover acknowledge funding from the Alexander von Humboldt Foundation within the framework of the Sofja Kovalevskaja Award, endowed by the German Federal Ministry of and Research, and the Technical University of Munich—Institute for Advanced Study, funded by the German Excellence Initiative and the European Union Seventh Framework Programme under Grant Agreement No. 291763. D.T. thanks Science Foundation Ireland (SFI) for support (awards Grant Numbers 15/CDA/3491 and 12/RC/2275_P2) and for computing resources at the SFI/Higher Education Authority Irish Centre for High-End Computing (ICHEC). Finally, we also acknowledge support from the U.S. National Science Foundation (Grant No. ECCS#1916874). The authors gratefully acknowledge the Gauss Centre for Supercomputing e.V. (www.gauss-centre.eu) for funding this project by providing computing time through the John von Neumann Institute for Computing (NIC) on the GCS Supercomputer JUWELS at Jülich Supercomputing Centre (JSC).

## Author contributions

X.C., D.A.E., and C.A.N. conceived and designed the project. X.C. synthesized the compounds, performed electrical characterizations, and associated data analysis; B.K. and D.A.E. performed the DFT calculations; F.A., C.N., and E.d.B. performed the Landauer model analysis; X.C. and X.Y. conducted the XPS and UPS measurements; D.T. conducted the molecular dynamics; all the authors discussed the results and prepared the manuscript.

## Competing interests

The authors declare no competing interests.
