## [Peer Review File · Nature Communications]

Reviewers' Comments:

Reviewer #1:

Remarks to the Author:

The paper describes molecular junctions whose tunneling decay coefficient can be controlled by changing the chemical contact to the top electrodes. The study attempts to explain how polarizable contacts changes the electrostatic potential profile of the tunneling barrier. This referee finds the findings are sound but incremental. The topic is not new and several studies report similar findings though the interpretations of the data are different. See for example the work of Lambert and co-workers, *Nanoscale*, 2018, 10, 3060-3067, which reports that the tunneling decay coefficient can be changed by the terminal group. Given the study is too specialized, I recommend publication in a more specialized journal.

Reviewer #2:

Remarks to the Author:

A single atom change turns insulating saturated wires into molecular conductors

Summary: In this work, Chen et al. have shown an extensive study on a series of alkanethiols (CnSH) terminating with X = H, F, Br, Cl, and I at the SAM//Ga2O3/EGaIn interface. The authors synthesized the relevant compounds and performed J-V characterization in AgTS/SAM//Ga2O3/EGaIn junctions to calculate beta values of CnSH with different X substitutions showing that the beta value can be successfully manipulated from 0.75 to 0.25 angstrom⁻¹. The SAMs were characterized using ARXPS, impedance spectroscopy, and supported by DFT and MD simulations. The authors have characterized the SAMs thoroughly and provided valuable insights into rates of tunneling charge transport, highlighting the role of dielectric constants, which could be crucial to the further development of useful molecular tunneling junctions. This work should be published after the authors address some issues contextualizing it (see major comments).

Major comments:

The authors should discuss and comment on in either the introduction or the discussion section, the several works from the Whitesides group and others where functionalization of alkanethiol did not affect the charge transport behaviour of alkanethiol SAMs. (*ACS Nano* 2015, 9, 1471-1477; *Angew. Chem. Int. Ed.* 2012, 51, 4658-4661; *J. Am. Chem. Soc.* 2014, 136, 16-19; *ACS Nano* 2018, 12, 10221-10230) While the standard, rectangular electrostatic barrier model is clearly an incomplete description, there are a surprising number of cases where it correctly predicts tunneling decay coefficients and their relative insensitive to groups at the SAM-electrode interface.

While the authors have clearly identified series of molecules for which the polarizability at that interface affects Beta, for this observation to be useful for the design of future molecules and experiments, it must also be able to explain the considerable number of cases where other, polarizable groups do not seem to have the same effect. This comparison is particularly important given that the authors have already published the observation that halogen atoms increase the rate of tunneling charge transport across alkane thiols in which they characterized the dielectric constants (*Adv. Mater.* 2015, 27, 6689-6695), nullifying the claim that the present work is the first report of using halogens to affect tunneling charge transport without modifying molecular backbones; what is new and interesting in the present work is the effect on the length-dependence.

The first paragraph of the Discussion begins with the assertion that it is reasonable that DFT predicts no change in dielectric constant across the series, but in *Adv. Mater.* 2015, 27, 6689-6695 the authors use the van der Waals DFT-D2 method to compute the dielectric constant and

find it in agreement with experimental values. Why do they now use a method that disagrees with experiment? It appears that the DFT calculations in the current work serve only to disagree with experiment (Fig 5b). The Discussion is not very clear as to whether any DFT values are used in the Landauer model or if all parameters were derived from fits. In the latter case, what is the purpose of the DFT calculations?

Minor comments:

Were the variable temperature measurements performed with conical tips? If not, was the junction area the same and what impact would leakage currents have on those measurements. Could it conceal a thermally activated component of charge transport? The schematic in Figure 1B suggests that iodine-terminated SAMs should rectify by the same thermally activated mechanism as other molecules with highly localized HOMOs near E_f .

The following sentences are repeated twice in the main text, one of which should either be removed or rewritten:

Lines 284 & 300 (similar)

Lines 319 & 320 (exact repeats)

In equation 6, the symbol '#' should be removed before the label '(6)'.

In line number 332, the description of γ_L and γ_R should be removed as it is already specified in line number 324.

In supporting information, line 453, authors specify that 100 mL ethanol was used to rinse every SAM. Is there a reason to use such high volumes? Usually, 3-5 mL of ethanol should be sufficient. Please either provide reasoning or correct the description if this is a typo.

Authors should provide further details on geometries of systems on which the DFT simulations were performed, such as, the xyz coordinates of at least one corresponding Ag-(CH₂)₁₄-X systems

Reviewer #3:

Remarks to the Author:

In this manuscript, the authors studied the tunneling behavior of a series of molecular junctions, S(CH₂)_nX, with n=10 to 18 (all even numbers), and X=H, F, Cl, Br, and I. The authors reported that by changing the identity of X, the decay coefficient, β , can change dramatically. The authors analyzed various aspects of the transport mechanism to explain this phenomenon. Overall the findings are interesting to the community, and the paper is generally well written. However, I found a few key things that are not thoroughly discussed, as I detail below. I think these points should be addressed before I can recommend publication.

(1) The authors only considered even numbers for n. I recognize that the authors mentioned the even-odd effect that some of the current authors discussed in Ref. 55. However, how does that effect relate to the present discussions in this work? In particular, does one expect the same qualitative behavior if one only focuses on the odd series? What would change and what would not if one considers all n, both even and odd?

(2) Around the discussion of Fig. 4, the authors used R_c , R_{SAM} and C_{SAM} (directly related to ϵ_r via Eq. 3) to understand the phenomenon. How are these quantities exactly defined and derived from experiments/calculations? I notice they are schematically defined in Fig. 1, but a rigorous definition and explanation of their physical meanings are still missing. I understand that they might have been defined elsewhere, but given their importance in the discussion, I believe brief comments in this paper are necessary;

(3) Again regarding the physical meanings of the above quantities. Are they designed to separate the effect of local binding motif, X, and the molecular backbone (which depends on n)? If this is the case, I doubt this separation does its job: I can understand the decreasing of R_c as shown in Fig. 4a, but why does the R_{SAM} physically decrease as a function of X as well, as shown in Table

S8? In Line 235, it is suggested that the impedance measurement leads to the same values as $J(V)$ measurement in Fig. 1. I can see the β agrees but not quite the J_0 value. It would be great to perform a direct comparison for one chosen system, between the measured $J(V)$ curve and the calculated $J(V)$ curve from quantities derived from the impedance measurement, i.e., using the circuit as shown in Fig. 1 to calculate $J(V)$. Is this possible and meaningful?

(4) In the DFT calculations, it seems to me that the top electrode (EGaIn/GaOx) is not included, as shown in Figs. S22 and S23. Does that affect the trends and explanations? Furthermore, since the key purpose of this paper is to explain the β change, which is related to trends as n increases. I unfortunately did not find any DFT results as a function of n for a fixed X . Ideally, it would make the argument stronger if the calculations can be performed for two X , say H and I, with varying n . In this way, one can derive qualitative trends (may not be in quantitative agreement with experiments, as the authors explained), i.e., whether the β is higher or lower when X is changed;

(5) In Fig. 6, again, I did not find any n dependence, which is central to the stated key findings of this paper (β change). I understand that Table S9 showed that ϵ_r , the quantity in Fig. 6f, does not significantly change as n . How about the quantities shown in Figs. 6b and 6c? Also for Fig. 6d, for which n is this figure plotted? I guess that the ΔE_{ME} and Γ might change as a function of n for a fixed X . If this is the case, I would like to see discussions along this line; if this is not the case, I would like to understand why. Without the n dependence here, I feel that a direct analysis on the β change based on the Landauer formula is missing;

(6) It looks to me that the central conclusion is that X modulates β via changing ϵ_r . However in Line 419-420, the authors mentioned that this was not reproduced in DFT calculations. Could it be that the definitions for the calculated ϵ_r (how is this quantity precisely defined from the calculations? We need some discussions here) and the measured ϵ_r (defined via Eq. 3) are just different in the sense that they microscopically describe different things? To resolve this, a careful discussion on the origin of the definition of C_{SAM} , used to define ϵ_r experimentally in Eq. 3, is necessary. This is a point I already listed above in (2), but I think it is worth being mentioned again here when considering the discrepancy between the computational and experimental determination of ϵ_r .

Apart from the above comments, there is one suggestion:

- I believe it is not quite appropriate to list β in Table 1 with the $n=14$ lines, as β reflects the change as a function of n . Based on Fig. 3c, how about plotting β as a function of X , with error bars? This is similar to Fig. 6f, but without the ϵ_r (which is a deduced quantity).

Please Note: The original comments by reviewer are in black color, and the responses and changes to manuscript are in blue.

REVIEWER COMMENTS

Response to Reviewer #1 (Remarks to the Author):

The paper describes molecular junctions whose tunneling decay coefficient can be controlled by changing the chemical contact to the top electrodes. The study attempts to explain how polarizable contacts changes the electrostatic potential profile of the tunneling barrier. This referee finds the findings are sound but incremental. The topic is not new and several studies report similar findings though the interpretations of the data are different. See for example the work of Lambert and co-workers, *Nanoscale*, 2018, 10, 3060-3067, which reports that the tunneling decay coefficient can be changed by the terminal group. Given the study is too specialized, I recommend publication in a more specialized journal.

Response:

We kindly thank the reviewer for reading our manuscript. We respectfully disagree with the reviewer as the topic is certainly novel as has been recognized by the other 2 reviewers who both support publication after minor revision with Reviewer #2 stating that our work "...**could be crucial to the further development of useful molecular tunneling junctions**". The charge transport mechanisms in molecular junctions and the associated length-dependent properties, such as, β , are still not fully quantitatively understood, and the role of dielectric effects remains largely unexplored. Understanding of the dielectric behavior of molecular junctions is thus still in its infancy. We show, for the first time, that there is a relation between β and ϵ , which we believe will motivate ample further scientific investigations.

Our work stands in stark contrast to the work by Lambert and co-works (*Nanoscale*, 2018, 10, 3060-3067) for the following reasons:

- 1) **Their work does not study nor mention the role of dielectric effects.**
- 2) **Their work studies SAMs with different backbones.** They investigated molecules of the form $An-(CH_2)_n-[FG]-(CH_2)_n-An$ with the functional group $FG =$ phenyl, viologen, or α -terthiophene, linked by methylene spacers to anchoring (An) groups of thiol or thiomethyl.
- 3) **Their work does not describe the role of the terminal group.** Instead the authors claim the anchoring groups provide low energy states that can facilitate tunneling rates *involving* the functional groups in the middle of the molecule. In other words, they claim that new interface states (which they called "gateway" states) induced by the Au-S bond are involved in tunneling *via* FG.

Furthermore, we would like to mention that the findings by Lambert and co-workers stand in sharp contrast to a recent study by Frisbie and co-workers (*Chem. Sci.*, 2018, 9, 4456-4467) who pointed out that the "gateway state" provided by the Au-S is insignificant due to opposing Stark effects. In addition, experimental UPS spectra of $Au-S(CH_2)_nCH_3$ SAMs show that the feature associated with the Au-S bond is barely measurable because of the strong hybridization of the S with the metal electrode (e.g., Frisbie and co-workers *J. Am. Chem. Soc.* 2019, 141, 18182–18192, and also in our calculations the PDOS associated with Au-S bond is minor and it is also not visible in our experimental UPS data).

These two contradicting papers and results highlight the importance of using optimized molecule-electrode contacts to ensure the molecular states remain localized on the molecule as we have done in this work: the EGaIn top-contact only interacts weakly with the SAM allowing us to study the effects of the terminal group in unprecedented detail. Such experiments cannot be conducted in break junctions as was done by Lambert et al. since with that method requires that the molecule must form relatively strong contacts with both electrodes.

Changes to the manuscript:

On Page 4, starting from Line 76, we added the following sentences to discuss the examples of Lambert and co-workers and Frisbie and co-workers which relate β to the molecular backbone and the molecule—electrode coupling strength.

“Lambert and co-workers were able to tune the β value between 0.06 and 0.39 \AA^{-1} in Au-S(CH₂)_nFG(CH₂)_nS-Au junctions with a functional group FG = α -terthiophene, phenyl, or viologen.³⁶ They found that changing the anchoring group from dithiol to dithiolmethyl for FG = phenyl resulted in an increase of the β value from 0.14 to 0.50 \AA^{-1} from which they concluded that localized states on the Au-S bond are involved in tunnelling along the FG units. In contrast, Frisbie and co-workers found that β values are similar for Au-S(CH₂)_nCH₃//Au and Au-S(CH₂)_nS-Au junctions, implying that localized states on the Au-S bond are not important for tuning β (but note that they still significantly affect the contact resistance).³⁷ Frisbie and co-workers suggested that Stark effects are important to consider as they can cancel the potential effects of localized anchoring group-electrode states.^{38,39} Indeed, strong Au-S interaction results in severe broadening of the molecular states and therefore the Au-S states only occur as weak features in valence band spectra of aliphatic SAMs³⁸ (as also observed in the present study), highlighting the need to optimize the molecule-electrode interaction strength such that the molecular states remain localized in the molecule. Here, we use junctions of the form Ag-S(CH₂)_nX//EGaIn ($n = 10, 12, 14, 16, \text{ or } 18$, and X = H, F, Cl, Br, or I) where the weak interaction between the top electrode and the SAM allows us to investigate in detail how the terminal group X affects the tunnelling rates across the junctions.”

Reviewer #2 (Remarks to the Author):

A single atom change turns insulating saturated wires into molecular conductors

Summary: In this work, Chen et al. have shown an extensive study on a series of alkanethiols (C_nSH) terminating with X = H, F, Br, Cl, and I at the SAM//Ga₂O₃/EGaIn interface. The authors synthesized the relevant compounds and performed J-V characterization in AgTS/SAM//Ga₂O₃/EGaIn junctions to calculate beta values of C_nSH with different X substitutions showing that the beta value can be successfully manipulated from 0.75 to 0.25 \AA^{-1} . The SAMs were characterized using ARXPS, impedance spectroscopy, and supported by DFT and MD simulations. The authors have characterized the SAMs thoroughly and provided valuable insights into rates of tunneling charge transport, highlighting the role of dielectric constants, which could be crucial to the further development of useful molecular tunneling junctions. This work should be published after the authors address some issues contextualizing it (see major comments).

Response: We kindly thank the reviewer for appreciating the importance of our work. The reviewer made constructive comments and suggestions which we have addressed in our revision and in response we have made substantial changes to our manuscript which helped us to improve discussions where needed.

Major comments:

Comment 1

The authors should discuss and comment on in either the introduction or the discussion section, the several works from the Whitesides group and others where functionalization of alkanethiol did not affect the charge transport behaviour of alkanethiol SAMs. (ACS Nano 2015, 9, 1471–1477; Angew. Chem. Int. Ed. 2012, 51, 4658–4661; J. Am. Chem. Soc. 2014, 136, 16–19; ACS Nano 2018, 12, 10221–10230) While the standard, rectangular electrostatic barrier model is clearly an incomplete description, there are a surprising number of cases where it correctly predicts tunneling decay coefficients and their relative insensitive to groups at the SAM-electrode interface.

Response: The reviewer makes a valid point and our results appear at first glance to contradict those reported by the Whitesides group. However, the Whiteside group studied many different functionalities but did not measure transport as a function of the alkyl chain lengths. Hence, any potential change in β has simply not been studied until now. Besides, the Whitesides group uses a modified EGaIn technique where the SAM//EGaIn top contact formation involves a much larger contact area than what we create. Therefore, they run a far greater risk of probing defects than we do.

Changes to the manuscript:

On Page 5, Line 109, we added several sentences discussing the previous related works:

“For instance, Whitesides and co-workers reported that the charge transport rates in metal-S-(CH₂)_nFG//EGaIn junctions with aliphatic SAMs are independent of FG with FG being terminal aromatic groups,⁴⁹ polar groups,⁵⁰ ionic and/or hydrogen bonding groups,⁵¹ or halogen atoms,⁴⁷ and concluded that changes in terminal group does not affect the charge transport rates. In these studies they used large junction areas of >1000 μm^2 , but we have shown that such large junctions are prone to defects masking molecular effects and that, for EGaIn-based methods, stable junctions that are dominated by molecular effects should have an area of 300-500 μm^2 .⁵² Indeed, the Whitesides group could reproduce our results and also found a factor of 600 in the charge transport rates when X=H was replaced with X=Br when small junctions were used.^{47,}”

Comment 2

While the authors have clearly identified series of molecules for which the polarizability at that interface affects Beta, for this observation to be useful for the design of future molecules and experiments, it must also be able to explain the considerable number of cases where other, polarizable groups do not seem to have the same effect. This comparison is particularly important given that the authors have already published the observation that halogen atoms increase the rate of tunneling charge transport across alkane thiols in which they characterized the dielectric constants (Adv. Mater. 2015, 27, 6689–6695), nullifying the claim that the present work is the first report of using halogens to affect tunneling charge transport without modifying molecular backbones; what is new and interesting in the present work is the effect on the length-dependence.

Response: Indeed, the novelty we report here is the length dependence which firmly establishes that β is changed (in addition to the experimentally determined correlation with the dielectric constant). We believe we have addressed this point sufficiently in response to the previous comment by the same reviewer.

Changes to the manuscript:

On Page 7, Line 153, we emphasize the new insight of this work:

“However, the evolution of J and ϵ_r with increasing molecular length and the corresponding β values for different X have so far not been studied.”

Comment 3

The first paragraph of the Discussion begins with the assertion that it is reasonable that DFT predicts no change in dielectric constant across the series, but in Adv. Mater. 2015, 27, 6689–6695 the authors use the van der Waals DFT-D2 method to compute the dielectric constant and find it in agreement with experimental values. Why do they now use a method that disagrees with experiment? It appears that the DFT calculations in the current work serve only to disagree with experiment (Fig 5b). The Discussion is not very clear as to whether any DFT values are used in the Landauer model or if all parameters were derived from fits. In the latter case, what is purpose of the DFT calculations?

Response: First, about the dielectric constant we must stress that unfortunately in the paper Adv. Mater. 2015, 27, 6689–6695 the DFT calculations were incorrect and agreed with experiment due to error cancellation, which was not realized at the time. Note that none of the authors responsible for the DFT calculations in this work was involved in the previous study. Thus, prior to submitting the present manuscript we performed extensive tests with multiple DFT codes to ensure the dielectric constants we obtained are reliable. Two results are important here:

- 1.) Including van der Waals interactions does not significantly change the dielectric constant obtained from DFT.
- 2.) Including dipole corrections is of utmost importance to obtain reliable dielectric constants. In short, calculating a 2D slab or thin film in a 3D periodic DFT code (such as VASP or SIESTA) requires to compensate for differences in the vacuum potentials on both sides of the slab/film to avoid spurious electrostatic effects. This is known as “dipole correction” and MUST be included in any periodic calculation for polar unit cells. However, in the study mentioned by the reviewer, the calculations were performed with a SIESTA version (3.2pl4) which at the time did not support the simultaneous application of dipole correction and external electric field. Therefore, the dipole moments were significantly overestimated, the dielectric constants reported previously were incorrect, and disagree with our current study. While our new theoretical data do not agree with the experimental findings for the dielectric constant (for various interesting physical reasons discussed in the paper), we are certain that the here the physics of the applied method is correct.

Regarding the purpose of our DFT calculations: they confirm the microscopic origin of the trends observed in the experiments. Our DFT calculations reveal that the HOMO and LUMO levels are shifted as a function of terminal group X. This can explain the change of the tunneling barrier as a function of X and, thus, the increase of conductivity when going from X=H to X=I. Furthermore, following the reviewer’s comment we used the energies of the HOMO levels obtained from DFT in the Landauer fits to model the height of the tunneling barrier. Note that for the range of applied bias scanned in these experiments, the junction does not enter into resonance (i.e., the maximum applied bias is not sufficient to place the molecular orbital into the conduction window). In this case, it is not possible to

unambiguously determine the orbital energies, since they are totally free fitting parameters competing with the tunneling rates (i.e., similar current results can be obtained with different combinations of these two fitting parameters). Therefore, we have chosen to use the values extracted from DFT in the revised version, although energy renormalization effects in two-terminal junctions are known to commonly decrease the orbital energies substantially compared to those obtained by DFT analysis and spectroscopic characterization in SAM formed on surfaces. Nonetheless, the main result shown by the Landauer fittings illustrates that both the orbital energy and the tunneling rates change as a function of the termination atom, in agreement with DFT and spectroscopic results.

The comment by the reviewer made us reevaluate our experimental and theoretical data for the work functions of all systems. We reanalyzed the experimental data and pursued extensive additional DFT modelling using a more realistic structural motif in which 4 molecules per cell form a herringbone pattern as discussed for alkylchains in the literature. This led to corrections of the experimental work function values, since there had been a mistake in the UPS data transfer process, the Fermi edges were not well-aligned and the work function values were not calibrated against a reference clean Ag. The new Fermi edges and work functions are plotted against clean Ag and shown in the following figure for the referee. It also led to some small changes in the DFT-calculated data. Taken together, the agreement between the experimental work function values and the theoretical predictions is much better now.

Changes to the manuscript:

On Page 11, we deleted Lines 256-258:

“For Ag-S(CH₂)₁₄X, we find significantly different tilting angles with respect to the surface normal, which span the range of 14.7° (X=H) to 32.8° (X=Cl). Note that these calculated interface geometries are the result of a 0 K optimization of a single-molecule unit cell.”

On Page 11, Line 254, we changed the sentence from

“we performed first-principles calculations based on DFT using the VASP code⁵⁶”

to

“we performed first-principles calculations based on DFT using the VASP code⁵⁶ and a $3 \times 2 \sqrt{3}$ Ag surface unit cell containing four molecules arranged in a herringbone pattern”

On page S46, Line 678, we added:

“We used a $3 \times 2\sqrt{3}$ Ag surface unit cell with a PBE optimized lattice constant of 4.14 Å and four molecules arranged in a herringbone pattern per unit cell (see Supplementary Fig.24). Similar structures have been reported for alkyl-thiol SAMs on Ag and Au.²² Note that we performed calculations on different herringbone structures (e.g., different relative orientation of the molecules), all of which are very close in energy after structural optimizations. Hence, we chose the structure for which the work-functions were closest to the experimental values.”

Supplementary Fig. 24. Top view of herringbone structure for Ag-S(CH₂)₁₄-H. The black solid lines indicate the cell (four cells are shown here to visualize the herringbone structure).

On Page 11, Line 261, we changed the sentence from

“it can be seen that the agreement for X=H is excellent (Fig. 5b), while both the absolute values and to a lesser degree also the chemical trends agree not as favourably for the halogen-substituted systems.”

to

“it can be seen that the agreement is good for all terminations except X=F (Fig. 5b).”

In order to prevent misunderstandings, on Page 13 Line 306 we changed the original sentence:

“We note that the physical significance and so the predictive power of periodic DFT calculations previously reported by some of us⁴⁶ were limited by simulation artefacts of uncompensated dipoles in the unit cell which may have created a spurious correlation with measured ϵ_r values.”

To

“We note that previously reported DFT calculations of ϵ_r of the HS(CH₂)₁₁X SAMs⁴⁵ were incorrect due to simulation artefacts of uncompensated dipoles in the unit cell, which created a spurious correlation with experimentally measured ϵ_r values.”

On Page S27 Line 492, we added the sentence:

“The Fermi edges and work function values were calibrated against a reference, clean Ag surface ($\Phi = 4.26$ eV).”

On Page 29, Table 1, we updated the experimental and theoretical work functions and $\epsilon_{\text{DFT-vdW}}$:

Table 1. Summary of properties of the Ag-S(CH₂)₁₄X and Ag-S(CH₂)_nBr SAMs

X and n	$\Psi_{\text{SAM,XPS}}$ (nmol/cm ²) ^a	$d_{\text{SAM,XPS}}$ (Å)	$d_{\text{SAM,MD}}$ (Å)	$E_{\text{mol,MD}}$ (eV)	Φ_{SECO} (eV) ^b	Φ_{DFT} (eV)	ϵ_r	$\epsilon_{\text{DFT-vdW}}$
n = 14, X = H	0.74	18	20.4±0.5	-1.8±0.1	3.98	3.47	2.9±0.3	2.2
n = 14, X = F	1.0	21	21.1±0.3	-2.0±0.1	4.43	5.44	2.5±0.6	2.1
n = 14, X = Cl	0.86	21	21.5±0.3	-2.2±0.2	5.02	5.25	3.0±0.2	2.2
n = 14, X = Br	1.1	20	21.7±0.3	-2.4±0.2	4.77	5.18	4.7±0.9	2.3
n = 14, X = I	1.2	21	21.8±0.3	-2.2±0.1	4.68	4.96	8.9±1.6	2.4
n = 10, X = Br	1.0	15	16.2±0.4	-1.7±0.1	4.62	-	4.4±0.4	-
n = 18, X = Br	1.1	29	26.8±0.3	-3.0±0.2	4.66	-	4.6±0.2	-

On Page 28, we updated Figure 5 with new experimental and DFT work functions.

Minor comments:

Comment 4

Were the variable temperature measurements performed with conical tips? If not, was the junction area the same and what impact would leakage currents have on those measurements. Could it conceal a thermally activated component of charge transport? The schematic in Figure 1B suggests that iodine-terminated SAMs should rectify by the same thermally activated mechanism as other molecules with highly localized HOMOs near E_f .

Response: The temperature dependent measurements were performed with EGaIn confined in a through-hole in microchannels fabricated in PDMS (Wan et al. *Adv. Funct. Mater.* 2014, 24, 4442-4456; Suchand et al. *Nanoscale* 2015, 7, 12061-12067). The junction area is about 960 μm^2 which is about 2.5 times larger than that of EGaIn tips (350-500 μm^2). We only used

junctions that had their $J(V)$ characteristics within one log-standard deviation ($\sigma_{\log,G}$) of the log-average $J(V)$ obtained from the EGaIn tip junctions. From our previous reports (e.g., Chen et al. *Nat. Nanotechnol.* 2017, 12, 797-803; Yuan et al. *Nat. Nanotechnol.* 2018, 13, 322-329), using this kind of PDMS device as the top electrode does not conceal a thermally activated component of charge transport. We note that the molecules we studied here are not redox-active and that even for the I-terminated SAMs the HOMO is still ~ 1.5 eV away from the E_F (see Figures 5c and 5d) from DFT calculations. To investigate the possibility of rectification, we would have to subject the junctions to very large bias windows ($> \pm 2$ V), well above the breakdown voltage of our junctions (*Adv. Funct. Mater* 2018, 1801710).

Changes to the manuscript:

On Page 9, Line 206, we expanded this statement

“We measured the $J(V)$ characteristics as a function of temperature, T in K, of Ag-S(CH₂)₁₄X//GaO_x/EGaIn junctions for all X.”

to include

“... using top electrode of EGaIn confined in a microfluidic network in polydimethylsiloxane (PDMS) following a previous reported method⁴⁰ (see Supplementary Section 6 for details).”

To Section S6, Page S38, Line 589, we added the following:

“The junction area of the devices with the EGaIn stabilized in microchannels in PDMS was 960 μm^2 and the junction area of the cone-shaped tip devices was 350-500 μm^2 . We only used junctions with $J(V)$ characteristics within one log-standard deviation ($\sigma_{\log,G}$) of that obtained from the cone-shaped tip junctions in our $J(V,T)$ measurements to ensure that the differences in junctions area did not cause any adverse effects due to leakage currents.^{15,16}”

On Page 12, Line 278, we inserted the sentences explaining why we did not see rectification in junctions.

“The X groups are not redox-active, and even for X = I the HOMO is still ~ 1.5 eV below E_F (Figs. 5c and 5d). Hence, the HOMO cannot enter the applied bias window of ± 0.5 V (note the molecules with $n = 10$ tend to break down at higher voltages)⁶¹.”

Comment 5

The following sentences are repeated twice in the main text, one of which should either be removed or rewritten:

Lines 284 & 300 (similar)

Changes to the manuscript:

We thank the reviewer for pointing out the oversights (comments 5-8).

On previous Page 16 Line 300, we deleted the repeated sentence: “Our theoretical finding that X functionalization does not dramatically alter ϵ_r of the SAM is in line with electrostatic considerations discussed in previous theoretical work.^{42,43,44}”

Comment 6

Lines 319 & 320 (exact repeats)

Changes to the manuscript:

On previous Page 17 Line 320, we deleted the repeated sentence: “The single-level Landauer model is also frequently used to model the current flowing across molecules junctions.”⁶⁶”

Comment 7

In equation 6, the symbol ‘#’ should be removed before the label ‘(6)’.

Changes to the manuscript:

In equation 6, the symbol “#” is removed.

Comment 8

In line number 332, the description of γ_L and γ_R should be removed as it is already specified in line number 324.

Changes to the manuscript:

On previous Line 332, we removed: “, where $\gamma_{L,R}$ are the electron tunneling rates between the molecule and the respective electrodes”.

Comment 9

In supporting information, line 453, authors specify that 100 mL ethanol was used to rinse every SAM. Is there a reason to use such high volumes? Usually, 3-5 mL of ethanol should be sufficient. Please either provide reasoning or correct the description if this is a type-O.

Response: We used a wash bottle to spray the ethanol to clean the SAM substrate for about 20 s to remove the residuals and physisorbed components. This is why a large amount of ethanol (~100 mL) was used. From our experience with XPS, CV, and other characterizations of several different types of SAMs (Nerngchanmng et al. *Nat. Nanotechnol.* 2013, 8, 113-118; Chen et al. *Nat. Nanotechnol.* 2017, 12, 797-803; Yuan et al. *Nat. Nanotechnol.* 2018, 13, 322-329), this method provides clean SAM surfaces.

On Page S24, Line 459, we changed the original sentences from

“The Ag substrates were left in the solutions for 3 h after which they were rinsed with 100 mL ethanol, and dried with N₂.”

to

“The Ag substrates were left in the solutions for 3 h after which they were rinsed with approximately 100 mL ethanol using a wash bottle and dried with N₂ flow.”

Comment 10

Authors should provide further details on geometries of systems on which the DFT simulations were performed, such as, the xyz coordinates of at least one corresponding Ag-(CH₂)₁₄-X systems

Response: We agree with the reviewer and added one of the geometries to the SI.

Changes to the manuscript:

Starting on Page S49, we added the POSCAR file containing the xyz coordinates of the relaxed structure of Ag-S(CH₂)₁₄H.

Reviewer #3 (Remarks to the Author):

In this manuscript, the authors studied the tunneling behavior of a series of molecular junctions, $S(\text{CH}_2)_n\text{X}$, with $n=10$ to 18 (all even numbers), and $\text{X}=\text{H}, \text{F}, \text{Cl}, \text{Br},$ and I . The authors reported that by changing the identity of X , the decay coefficient, β , can change dramatically. The authors analyzed various aspects of the transport mechanism to explain this phenomenon. Overall the findings are interesting to the community, and the paper is generally well written. However, I found a few key things that are not thoroughly discussed, as I detail below. I think these points should be addressed before I can recommend publication.

Response: We appreciate the reviewer for pointing out the importance of our manuscript and supporting its publication. We also thank the reviewer for the constructive comments and suggestions, which helped us to improve the clarity of our manuscript considerably.

Comment 1

(1) The authors only considered even numbers for n . I recognize that the authors mentioned the even-odd effect that some of the current authors discussed in Ref. 55. However, how does that effect relate to the present discussions in this work? In particular, does one expect the same qualitative behavior if one only focuses on the odd series? What would change and what would not if one considers all n , both even and odd?

Response: Impedance studies on molecular junctions are still rare. The only reason why we mentioned the odd-even effects in Ref. 55 was to place the results of the current manuscript in perspective. We realize now that this comparison caused unintended confusion as in ref 55 only unfunctionalized aliphatic SAMs were studied with both odd and even values of n , whilst in the current study, as the reviewer mentions, we only report even- n SAMs. We apologize for the unintentional confusion caused and therefore we have now removed this comparison.

Changes to the manuscript:

On Page 10, Line 220, we changed the original sentences from

“This change in R_C is substantially larger than the odd-even effect in R_C which results in a $2.5 \text{ m}\Omega/\text{cm}^2$ modulation of R_C in $S(\text{CH}_2)_{n-1}\text{CH}_3$ SAMs⁵⁵ and indicates that Γ substantially increases as function of X .”

to

“This change in R_C indicates that Γ substantially increases as a function of X .”

Comment 2

(2) Around the discussion of Fig. 4, the authors used R_c , R_{SAM} and C_{SAM} (directly related to ϵ_r via Eq. 3) to understand the phenomenon. How are these quantities exactly defined and derived from experiments/calculations? I notice they are schematically defined in Fig. 1, but a rigorous definition and explanation of their physical meanings are still missing. I understand that they might have been defined elsewhere, but given their importance in the discussion, I believe brief comments in this paper are necessary;

Response: We agree that every paper should be self-contained. The value of each of these parameters is directly determined by fitting the equivalent circuit to the experimentally obtained impedance spectra where the physical meaning of R_{SAM} and C_{SAM} are given in Eqs.

2 and 3, and the physical interpretation of R_C is now stated on Page 6 Line 140. For the sake of completion, we have added to Section S7 the mathematical form of the equivalent circuit used to fit our data where R_C , R_{SAM} and C_{SAM} are precisely defined.

Changes to the manuscript:

On Page 6 Line 138, we inserted:

“The equivalent circuit and the associated physical meaning of each circuit component has been explicitly discussed in our previous work⁴⁰ (and is summarized in Supplementary Section S7). Briefly, the R_C includes the resistances of the contacts of the SAM with the top and bottom electrodes, and the resistance of the electrodes and wires connecting the junction with the electrometers. The SAM itself behaves as a capacitor (C_{SAM}) with associated resistance (R_{SAM}) as expressed in Eqs. 2 and 3.”

On Page 9, Line 214 we expanded the sentence:

“and the data were fitted to the equivalent circuit shown in Fig. 1a following a previously reported method⁴⁰ (see Supplementary Section S7 for details).”

On Page S39, Line 600, we added the explanations of how the values R_C , R_{SAM} and C_{SAM} are derived from the impedance data using the equivalent circuit in Figure 1a:

“Under the disturbance of a sinusoidal voltage (E , in V) with a frequency ω (in rad/s, $\omega = 2\pi f$, f is frequency, in Hz), the capacitor gives a capacitive reactance X_C of

$$X_C = 1/\omega C \tag{S1}$$

Therefore, the impedance is more than the pure resistance of the circuit because it includes all the components that contribute to the complex impedance (Z , in Ω) under alternating voltage E .¹⁸

$$E = E_0 \sin(\omega t) \tag{S2}$$

The output of alternating current I (in A) is expressed with phase shift ζ , in $^\circ$, as:

$$I = I_0 \sin(\omega t + \zeta) \tag{S3}$$

The complex impedance Z is expressed with the real part of Z (Z') and the imaginary part (Z''), and the modulus of the impedance ($|Z|$) is the output of the impedance device:

$$Z = \frac{E}{I} = Z' + jZ'' = |Z|e^{j\zeta} \tag{S4}$$

The molecular junctions can be modelled with the equivalent circuit as shown in Fig. 1a, for which Z is given as¹⁸

$$Z = \left(R_C + \frac{R_{SAM}}{1 + \omega^2 R_{SAM}^2 C_{SAM}^2} \right) - j \left(\frac{\omega C_{SAM} R_{SAM}^2}{1 + \omega^2 R_{SAM}^2 C_{SAM}^2} \right) \tag{S5}$$

The values of R_C , R_{SAM} and C_{SAM} were derived from Eq. S5, and their physical meanings are explained in the main text.”

Comment 3

Again regarding the physical meanings of the above quantities. Are they designed to separate the effect of local binding motif, X, and the molecular backbone (which depends on n)? If this is the case, I doubt this separation does its job: I can understand the decreasing of R_C as shown in Fig. 4a, but why does the R_{SAM} physically decrease as a function of X as well, as shown in Table S8? In Line 235, it is suggested that the impedance measurement leads to the same values as J(V) measurement in Fig. 1. I can see the β agrees but not quite the J_0 value. It would be great to perform a direct comparison for one chosen system, between the measured J(V) curve and the calculated J(V) curve from quantities derived from the

impedance measurement, i.e., using the circuit as shown in Fig. 1 to calculate $J(V)$. Is this possible and meaningful?

Response:

Indeed, Impedance Spectroscopy (IS) allows us to disentangle the contributions of the different components of the junctions to a level of detail that is not possible with DC analysis. The IS data show that the changes observed in the $J(V)$ measurements are mainly caused by changes in the intrinsic properties of the SAM (the R_{SAM} and C_{SAM}), and changes in the contact resistance only plays a minor role. The DFT calculations indicate that by changing X, the tunneling barrier height and shape changes, which partially explains the changes in β . Given the similarity between β obtained from the $J(V)$ and R_{SAM} data, we conclude that the major driver behind the changes in β is related to R_{SAM} and C_{SAM} . In other words, the changes in electrical behavior are molecular driven, and not primarily driven by changes in the molecule—electrode interfaces. This is also reflected in the changes in ϵ which is not an interface effect. However, our results also reveal a small contribution from changes in R_C which is reflected in changes in the coupling strength between the molecules and electrodes Γ (and of course Γ in turn also affects the tunneling barrier height, inherent to the Landauer theory). We have discussed this in the Discussion section (Page 16 Line 384-391), and added the short discussion on Page 10, Line 229. The reviewer is correct that while we can quantify the contributions of the different circuit components, IS does not directly reveal how these different circuit components are influenced by each other (for which we need DFT and Landauer modelling).

The reviewer also asks us to compare $R_{\text{SAM},0}$ with J_0 from DC measurements. The second question is to calculate the $J(V)$ curves using the equivalent circuit but for that the bias dependency of each circuit component has to be known. We reported that before and R_C and C_{SAM} are independent of bias and R_{SAM} vs. n follows the trends of $J(V)$ (Sangeeth et al.; *Nanoscale* **2015**, 7, 12061-12067) and therefore we have not repeated such measurements here (since the $J(V)$ curves follow typical tunneling behavior and can be fitted very well with the Landauer formula as discussed in the manuscript).

The relation between J_0 and $R_{\text{SAM},0}$ is now explained in detail as described below.

Changes to the manuscript:

We have already defined all circuit symbols in detail in response to the previous comment by the same reviewer. On Page 10, Line 223, we added a discussion for the change of R_{SAM} vs. n :

“Supplementary Table 9 shows the decrease of R_{SAM} with X which is mainly caused by lowering of δE_{ME} and increase of Γ (see the DFT section).”

On Page 10, Line 229, we changed the original sentences from

“Figure 4b shows the plot of $\log_{10}R_{\text{SAM}}$ vs. n along with a fit to Eq. 2 from which we extracted the values of $\beta = 0.41 \pm 0.03 \text{ \AA}^{-1}$ and $\log_{10}|J_0| = 2.0 \pm 0.2 \text{ A/cm}^2$ which are, within error, the same as those values determined with the $J(V)$ measurements (Fig. 3c and Table 1)”

to

“Figure 4c shows the plot of $\log_{10}R_{\text{SAM}}$ vs. n along with a fit to Eq. 2 from which we extracted the values of $\beta = 0.41 \pm 0.03 \text{ \AA}^{-1}$ and $\log_{10}R_{\text{SAM},0} = -2.0 \pm 0.2 \text{ \Omega/cm}^2$ (or $R_{\text{SAM},0} = 1.0 \times 10^{-2} \text{ \Omega/cm}^2$). The value of $R_{\text{SAM},0}$ is essentially equivalent to J_0 (defined in Eq. 1) derived from a current decay plot at 30 mV (since the sinusoidal perturbation

used in the impedance measurements was 30 mV). The value of J_0 at 30 mV is 5.1 ± 2.0 A/cm² and the $\beta = 0.46 \pm 0.03$ Å⁻¹ (Supplementary Fig. 17). $R_{\text{SAM},0} \approx V/J_0 = 0.59 \times 10^{-2}$ Ω/cm², which is within a factor of 2 of the value measured with IS ($R_{\text{SAM},0} = 1.0 \times 10^{-2}$ Ω/cm²). The contribution of R_C is minor since R_C is a parallel circuit element but it is included in J_0 . The β and $R_{\text{SAM},0}$ values are, within error, the same as the values determined with the $J(V)$ measurements.”

On Page 11 Line 247, we inserted:

“Moreover, even though we can quantify the contributions of different circuit components from impedance spectroscopy, how these components are influenced by each other are not directly revealed. Therefore, we referred to DFT and Landauer modelling for further explanations.”

On Supplementary Page S36, we added the plot of $\langle \log_{10}|J| \rangle_G$ vs. n for junctions of X = Br at DC bias of -0.030 V in Supplementary Fig. 17.

Supplementary Fig. 17. The plot of $\langle \log_{10}|J| \rangle_G$ vs. n for junctions of X = Br for $V = -30$ mV. The error bars represent the $\sigma_{\log,G}$, and the red line is fit to Eq. 1. The plot indicates $\beta = 0.46 \pm 0.03$ Å⁻¹ and $\log_{10}|J_0| = 0.71 \pm 0.20$ A/cm² ($J_0 = 5.1 \pm 2.0$ A/cm²).

Comment 4

(4) In the DFT calculations, it seems to me that the top electrode (EGaIn/GaOx) is not included, as shown in Figs. S22 and S23. Does that affect the trends and explanations? Furthermore, since the key purpose of this paper is to explain the β change, which is related to trends as n increases. I unfortunately did not find any DFT results as a function of n for a fixed X. Ideally, it would make the argument stronger if the calculations can be performed for two X, say H and I, with varying n . In this way, one can derive qualitative trends (may not be in quantitative agreement with experiments, as the authors explained), i.e., whether the β is higher or lower when X is changed;

Response: EGaIn/GaOx is a liquid metal covered with an oxide layer, of which the atomistic details of the structure are unknown. Thus, the modeling of EGaIn/GaOx is out of the scope of this work. In preliminary test calculations using Ag as second electrode at a distance where it only interacts weakly (via vdW interactions) with the SAM, we mainly found the 2nd electrode to cause a small constant shift down in energy of the molecular DOS. The trends and explanations are not affected by this shift.

Furthermore, the reviewer asks us to perform additional calculations with another value of n , for $X=I$ and H . We performed these additional calculations for $n=10$ and found that the comparison between $X=I$ and $X=H$ between all theoretical quantities reported in the original manuscript is very similar compared to the originally contained calculations of $n=14$. We added these additional data to the SI of the revised manuscript.

Changes to the manuscript:

On Page S49, we added the descriptions for the calculations for $n=10$ with $X=H$ and $X=I$ and Supplementary Fig. 27:

“Calculations for $n=10$, with $X=H$ and $X=I$:

The differences in the electronic structure between $Ag-(CH_2)_{10}-X$ and $Ag-(CH_2)_{14}-X$ are minor. While the dielectric constants are 0.1 lower for $n=10$ than for $n=14$, the work-functions are slightly (~ 0.05 eV) larger for the shorter chains. The differences in the DOS are of the same order of magnitude as in the work-functions (see Supplementary Fig. 27).

Supplementary Fig. 27. DOS projected onto the organic part of $Ag-(CH_2)_{10}-X$ and $Ag-(CH_2)_{14}-X$ SAMs.”

Comment 5

(5) In Fig. 6, again, I did not find any n dependence, which is central to the stated key findings of this paper (β change). I understand that Table S9 showed that ϵ_r , the quantity in Fig. 6f, does not significantly change as n . How about the quantities shown in Figs. 6b and 6c? Also for Fig. 6d, for which n is this figure plotted? I guess that the ΔE_{ME} and Γ might change as a function of n for a fixed X . If this is the case, I would like to see discussions along this line; if this is not the case, I would like to understand why. Without the n dependence here, I feel that a direct analysis on the β change based on the Landauer formula is missing;

Response: We are sorry that we did not indicate the n value in the Figure 6, which is $n = 14$.

From the contact resistance R_C from impedance, we did not see an obvious change of R_C vs n for $X = \text{Br}$. From our previous work (Jiang et al, *J. Am. Chem. Soc.* 2015, 137, 10659-10667; Jiang et al, *Nano Lett.* 2015, 15, 6643–6649; Suchand et al. *J. Am. Chem. Soc.* 2014, 136, 11134–11144), the R_C vs. n for $X = \text{H}$ also shows similar values. Therefore, the Γ should be similar for all n . In Landauer formula, there is no parameter for molecular thickness (n), which will be reflected in the transmission probability (T), a combination effect of the fit parameters.

In Comment 3 of Reviewer 2, we have discussed the Landauer formula fit in detail and they confirm the microscopic origin of the trends observed in the experiments. As described below we have refined our model and refitted the $J(V)$ data.

Changes to the manuscript:

On Page 13, Line 323, we added the Fermi functions to Landauer fit:

“The $f_L(E)$ and $f_R(E)$ are the Fermi functions representing the electronic occupation of the left and right electrodes, respectively, which are given by⁶⁵

$$f_{L,R}(E) = \frac{1}{1 + \exp\left[\frac{E \pm V}{k_B T}\right]} \quad (6)”$$

The discussion part is also update

from

“Good fittings to the data for all molecules (Fig. 6a and Table S11) are achieved for $T = 300$ K and the following common parameters are obtained across all molecules: $N = 50$, $\eta \in 0.48 \pm 0.02$, and $\sigma = 0.19$ eV. Figures 6b and 6c show the two parameters that vary across molecules: the energy δE_{ME} of the frontier orbital, which decreases on moving through the sequence H-F-Cl-Br-I (Fig. 6b), in agreement with the behaviour observed in spectroscopic data and determined by DFT calculations (although of lower values due to energy renormalization absent in our DFT calculations),”

to

“We accounted for the behaviour of a group of molecules by setting the number of such molecules fixed at $N=150$. As obviously seen in the above model, the current is directly dependent on $\gamma_L \times \gamma_R$. There is a trade-off between the Gaussian and the density of states which is in the shape of a Lorentzian centred at the energy level δE_{ME} . All the molecules appeared to be symmetric and what accounts for the difference in conductance is the terminal atom on the molecular unit. In this case, the ligands were $X=\text{H, F, Cl, Br, I}$. Therefore five different set of fittings were done for each X , fixing δE_{ME} to the values extracted from DFT and leaving γ_L, γ_R, η and σ as fitting parameters to obtain best fits to the data of junctions of Ag-S(CH₂)₁₄X//GaO_x/EGaIn (Supplementary Table 12). Figure 6a shows the fits of the theoretical model (orange lines) to the experimental data (symbols) for each S(CH₂)₁₄X molecule. Figures 6b and 6c show the two parameters that vary across molecules: the energy δE_{ME} of the frontier orbital (extracted from DFT), which decreases from 4.3 to 1.7 eV on moving through the sequence H-F-Cl-Br-I (Fig. 6b),”

Figure 6 has been replaced with the new fitting parameters:

On Page S58, we replaced the original Supplementary Table 12 with the new table:

Parameters	X=H	X=F	X=Cl	X=Br	X=I
Γ (meV)	0.497	0.529	1.74	5.99	14.4
η	0.470	0.460	0.450	0.480	0.480
δE_{ME} (eV)	4.300	3.600	2.600	2.200	1.700
σ (eV)	0.190	0.190	0.190	0.190	0.190

Comment 6

(6) It looks to me that the central conclusion is that X modulates β via changing ϵ_r . However in Line 419-420, the authors mentioned that this was not reproduced in DFT calculations. Could it be that the definitions for the calculated ϵ_r (how is this quantity precisely defined from the calculations? We need some discussions here) and the measured ϵ_r (defined via Eq. 3) are just different in the sense that they microscopically describe different things? To resolve this, a careful discussion on the origin of the definition of C_{SAM} , used to define ϵ_r experimentally in Eq. 3, is necessary. This is a point I already listed above in (2), but I think it is worth being mentioned again here when considering the discrepancy between the computational and experimental determination of ϵ_r .

Response: The definitions of R_C , R_{SAM} and C_{SAM} and explanation of the physical meaning are the same as the response to Comment 2. In DFT, ϵ_r is calculated from the change in the dipole moment induced by an applied static electric field. Hereby, the atomic positions are fixed to the equilibrium positions when the field is applied. Hence, ϵ_r from DFT is the instantaneous response of the electronic charge density to a static electric field. Most importantly, however, the DFT calculations probe the intrinsic dielectric properties of the isolated highly organized

SAM without contacts, in contrast to experiment, where contacts are always present. We already discussed this in detail on Pages 12-13 of the manuscript. Furthermore, on Page 13, Line 301-306, we discussed possible reasons for the discrepancy between the computational and experimental ϵ_r values. The discrepancy is not unexpected considering various previous theory papers, as we discussed on Page 12, Line 290-293, where we mentioned that “Briefly, in these studies it has been shown from electrostatic and DFT calculations that varying the molecular polarizability of the SAM-forming molecules does not result in significant changes of ϵ_r in the densely-packed conjugated SAMs due to depolarization effects arising from the neighbouring molecular dipoles in the SAM^{41, 42, 43}.”

Changes to the manuscript:

On Page 12, Line 284, we added:

“Hereby, ϵ_r is calculated from the change in the dipole moment induced by an applied static electric field, with the atomic positions fixed at their equilibrium positions. Thus, the ϵ_r obtained in such a manner represents the instantaneous response of the electronic charge density to a static electric field.”

Comment 7

Apart from the above comments, there is one suggestion:

- I believe it is not quite appropriate to list β in Table 1 with the $n=14$ lines, as β reflects the change as a function of n . Based on Fig. 3c, how about plotting β as a function of X , with error bars? This is similar to Fig. 6f, but without the ϵ_r (which is a deduced quantity).

Response: we agree it is not appropriate to put β values in the rows which have $n = 14$. We used a separate table (Supplementary Table 7) to indicate β and $\log_{10}|J_0|$ vs. X .

We agree to add the plot of β (with error bars) of vs. X which is a straightforward way to show the evolution of β . We show the plot on Page S36, Supplementary Fig. 18.

Changes to the manuscript:

On Page S37, we added a Supplementary Table 7:

Supplementary Table 7. Summary of the β and $\log_{10}|J_0|$ of junctions of Ag-S(CH₂)_nX//GaO_x/EGaIn junctions ± 0.5 V.

X	β (\AA^{-1})	$\log_{10} J_0 $ (A/cm ²)
H	0.75 \pm 0.04	2.5 \pm 0.1
F	0.70 \pm 0.02	2.6 \pm 0.3
Cl	0.60 \pm 0.03	2.6 \pm 0.2
Br	0.39 \pm 0.04	2.2 \pm 0.2
I	0.25 \pm 0.01	1.9 \pm 0.1

On Page S36, we added the plot of β (with error bars) vs. X , as shown below:

Supplementary Fig. 18. The plot of β vs. X. The error bars of β represent the standard errors from fitting to data to Eq. 1. The dashed red line is a guide to the eye.

On Page 9, Line 201, we added a sentence:

“Supplementary Fig. 18 shows the plot of β vs. X.”

Editorial changes:

The followings are editorial changes according to the Nature Communications Formatting Instructions. The editorial requests are resolved without scientific changes to the manuscript.

- 1) We named the article sections according to the Nature Communications formatting instructions.
- 2) We moved the second paragraph of original Introduction (Page 3) which discusses the results of current study to the last paragraph of the Introduction (Page 6). The original sentences, as shown below, on Page 6 was deleted due to duplicate:

“Here, we show that by introducing a single polarizable atom per molecule inside the junctions, the value of β can be reduced by a factor of 3 without changing the chemical structure of the backbone of the molecular wire. In terms of absolute values of J along long molecular wires of $\text{S}(\text{CH}_2)_{18}\text{X}$, the value of J increases by a factor of $10^{4.5}$ when X is changed from H to I. Our results demonstrate how an increasing polarizability of X changes the energy level alignment, the molecule–electrode coupling, and ε_r . These findings may stimulate new strategies for optimizing tunnelling rates across junctions.”
- 3) We placed all the figures and tables to the end of the manuscript, and added a short, standalone title for each figure.
- 4) We removed the inset from the original Figure 4b out and set it as a separate panel. So the Figure 4 is changed from original 3 panels to 4 panels now:
From

To

- 5) We reduced the number of references to below 70.
- 6) The citations of supplementary figures are in the correct format. For example: “Supplementary Fig. 1”.
- 7) The Editorial policy checklist form is also completed and updated.
- 8) We added the Data and Code availability sections.
- 9) All the corresponding authors have linked ORCID to the manuscript tracking system.
- 10) We also corrected the typos in the manuscript and Supplementary information.

Reviewers' Comments:

Reviewer #1:

Remarks to the Author:

NA

Reviewer #2:

Remarks to the Author:

Comment 1:

The authors addressed this point by pointing out that the junction area can affect the interpretation of subtle effects, for example that difference in results between the Whitesides group and theirs is because the former are more prone to probing defects because of big junction area ($>1000\mu\text{m}^2$), and therefore, do not see variation in tunneling transport with change in functional head groups. However, for variable-temperature measurements, the authors use micropore-based PDMS devices with junction area of $960\mu\text{m}^2$, i.e. bigger junction area as Whitesides', and observe change in J with X , demonstrating that junction area alone cannot be cause. This point should be addressed more forcefully, as this work is, at least in part, refuting earlier work.

Comment 2:

The statement: "To the best of our knowledge, this is the first example of tuning β across alkyl chains opening up new ways to control tunnelling rates in junctions." Is a bit of an overstatement. Something like: "Following our earlier research where we demonstrated that halogen atoms increase the tunneling charge transport and dielectric constant of alkane thiol SAMs,⁴⁶ in this study we further expand on the dependence of β with changing backbone length." Is more accurate.

Comment 3:

The change in pg 13 line 306: "We note that previously reported DFT calculations of ϵ_r of the HS(CH₂)₁₁X SAMs⁴⁵ were incorrect due to simulation artefacts of uncompensated dipoles in the unit cell, which created a spurious correlation with experimentally measured ϵ_r values." Shortened the explanation to a concise statement, but should now also include a more detailed explanation in the SI along the same lines as their detailed rebuttal to Comment 3 from the initial submission rebuttal.

Reviewer #3:

Remarks to the Author:

The authors have satisfactorily addressed all my questions and concerns from the first round of review in detail. Therefore I support publication as is.

We thank the reviewers for their comments on our manuscript. It seems all reviewers support publication of our manuscript in Nature Communications. In their previous report, reviewer 1 had raised concerns questioning the novelty of our work. We showed in our response that they were entirely unfounded. After our very detailed response and changes we made to the text, reviewer 1 did not provide any further comments, which in and of itself implies that our changes did satisfy the reviewer.

Reviewer 2 has already been highly supportive of our work in their previous report, to cite:

“The authors have characterized the SAMs thoroughly and provided valuable insights into rates of tunneling charge transport, highlighting the role of dielectric constants, which could be crucial to the further development of useful molecular tunneling junctions. **This work should be published** after the authors address some issues contextualizing it (see major comments).”

The remaining very minor comments by reviewer 2 are addressed below.

Reviewer 3 recommends publication as is.

Detailed response to the reviewer comments

Reviewer #1 (Remarks to the Author):

N/A

Response

None.

Reviewer #2 (Remarks to the Author):

Comment 1:

The authors addressed this point by pointing out that the junction area can affect the interpretation of subtle effects, for example that difference in results between the Whitesides group and theirs is because the former are more prone to probing defects because of big junction area ($>1000\mu\text{m}^2$), and therefore, do not see variation in tunneling transport with change in functional head groups. However, for variable-temperature measurements, the authors use micropore-based PDMS devices with junction area of $960\mu\text{m}^2$, i.e. bigger junction area as Whitesides', and observe change in J with X , demonstrating that junction area alone cannot be cause. This point should be addressed more forcefully, as this work is, at least in part, refuting earlier work.

Response

We thank the reviewer for raising a valid point, and we now make this point more forcefully in the revised manuscript (see Changes to the manuscript, below). Please note that our micropore devices are still smaller area than the junctions used by the Whitesides group (they use geometrical contact areas in the range of $1000\text{-}4300\mu\text{m}^2$ but they do not give exact

numbers in each paper). As we have clarified in the previous round of review, the Whitesides group can repeat our results once they also used small junction area. The reason is that the use of large junctions risks probing defects as we have discussed before in several works (as explained on page 3 of our original rebuttal, in response to Comment 1 by the same Reviewer). The micropore devices do give the same trend as the junctions made with small tips, which means that the micropore device junctions form more gentle contacts with the SAM than the large tips do. The large tips must be pushed against the SAM because those junctions have a larger area than what is expected from the tip radius. Although junctions at $\sim 1000 \mu\text{m}^2$ perform in general worse than small area junctions, a considerable fraction of the junctions show the same characteristics as small area junctions with a geometrical area of $350\text{-}500 \mu\text{m}^2$ (i.e., overlapping distributions as described in *ACS Appl. Mater. Interfaces* 2019, 11, 21018-21029). Therefore, we first used cone-tip junctions to determine the **averaged** $\log_{10}|J|$ and β , then we chose junctions obtained with the microfluidic devices with their characteristics within one log-standard deviation of the log-average for detailed studies.

Changes to the manuscript

To clarify ourselves, we added the following sentence, on page S30.

“The junctions with the EGaIn stabilized in a micropore (see below) have a relatively large geometrical contact area of $960 \mu\text{m}^2$, but this method does not suffer from the significant leakage currents found in the cone-shaped tip junctions, which arise due to the need to push the EGaIn tip against the SAM until the desired geometrical contact area is obtained. For junctions with the EGaIn stabilized in a through-hole in PDMS, we only used junctions with $J(V)$ characteristics within one log-standard deviation ($\sigma_{\log,G}$) of that obtained from the cone-shaped tip junctions in our *subsequent* measurements to ensure that the differences in junctions area did not cause any adverse effects due to leakage currents.^{14,15}”

In the original manuscript, we also state on page S38:

“We only used junctions with $J(V)$ characteristics within one log-standard deviation ($\sigma_{\log,G}$) of that obtained from the cone-shaped tip junctions in our $J(V,T)$ measurements to ensure that the differences in junctions area did not cause any adverse effects due to leakage currents.^{14,15}”

Comment 2:

The statement: “To the best of our knowledge, this is the first example of tuning β across alkyl chains opening up new ways to control tunnelling rates in junctions.” Is a bit of an overstatement. Something like: “Following our earlier research where we demonstrated that halogen atoms increase the tunneling charge transport and dielectric constant of alkane thiol SAMs,⁴⁶ in this study we further expand on the dependence of β with changing backbone length.” Is more accurate.

Response

We respectfully disagree with the reviewer, because to the best of our knowledge changing β across alkyl chains has not been reported before and that is the novelty of this work; in our previous work, we did not report values of β . We have already

addressed this issue in the previous round of review, and we wish to avoid increasing the length of our manuscript unnecessarily. If the reviewer can point us to specific works, we are more than happy to reword the mentioned statement in our manuscript.

Comment 3:

The change in pg 13 line 306: “We note that previously reported DFT calculations of ϵ_r of the HS(CH₂)₁₁X SAMs⁴⁵ were incorrect due to simulation artefacts of uncompensated dipoles in the unit cell, which created a spurious correlation with experimentally measured ϵ_r values.” Shortened the explanation to a concise statement, but should now also include a more detailed explanation in the SI along the same lines as their detailed rebuttal to Comment 3 from the initial submission rebuttal.

Response

We agree that this could be helpful and we now include additional explanations in the revised supporting information of the article.

Changes to the supporting information

On page S47 of the supporting information, we added:

“We note that unfortunately in Ref. 1 the DFT calculations were incorrect and happened to agree with experiment due to error cancellation, which was not realized at the time. The authors responsible for the DFT calculations in the present work (who were not involved in the previous study) found that the mistake was based on not including a dipole correction in this earlier work, which would have been of utmost importance to obtain reliable dielectric constants. In short, accurate electronic structure calculations of a polar 2D slab or thin film in a 3D simulation cell must correct for differences in the vacuum potentials on both sides of the slab or film to avoid spurious electrostatic effects. This is known as a “dipole correction” and must be included in periodic DFT calculations of such systems. However, in Ref. 1 this was not done and, therefore, the field-induced dipole moments were significantly overestimated. As a consequence, the dielectric constants reported previously were incorrect and disagree with the values reported in our current study.”

Reviewer #3 (Remarks to the Author):

The authors have satisfactorily addressed all my questions and concerns from the first round of review in detail. Therefore I support publication as is.

Response

We thank the reviewer!

Reviewers' Comments:

Reviewer #2:

Remarks to the Author:

The revised manuscript addresses all of the scientific concerns of the previous versions, but the author's assertion that manipulating Beta values of alkanes is an important breakthrough and departure from earlier work still seems over stated.

This work is very rigorous and the phenomenon is very well characterized, but fundamentally, Beta is proportional to the barrier height, so any change to the barrier height will change Beta. The claim in this work is that the authors are manipulating Beta for alkanes and are doing so by changing a single atom. But that atom is a halogen, which is a significant change. Hexane, 1-chlorohexane and 1-iodohexane have very different physical properties and reactivities. So to some extent the central claim is semantic, since one could just as easily subjectively characterize the addition of a ferrocene as a minor change that has a significant effect on an alkane.

Put another way: *reductio ad absurdum*, the claim is that manipulating Beta in alkanes without any synthetic modification is the underlying motivation. But that can be done by electrostatic gating, which was already demonstrated in 2009 (10.1038/nature08639).

Also, technically, this is not the first demonstration of manipulating Beta in alkanes:
10.1038/s41563-020-00876-2

Granted the effect is smaller, but it is also a single atom exerting a large effect on coupling, just at the opposite electrode. And it most certainly opens up new ways to control tunneling rates in junctions, since the effect on conductance is significant.

This is definitely interesting work, but the authors should nonetheless consider tempering their language a bit and highlight the magnitude of the effect on Beta, which is an interesting observation, rather than the novelty of the phenomenon itself, which leans on semantics.

Reviewer #4:

Remarks to the Author:

In my opinion, Xiaoping Chen and co-workers gathered a very broad and detailed collection of experimental and theoretical data utilizing various state-of-the-art investigation techniques to study in a very systematic manner the impact of the terminal group of alkane thiol SAMs on their charge transport characteristics. Considering the large number of different molecule types that were characterized (different chain length and different terminal groups) as well as the extension of the investigation methods in comparison to previous papers and the interpretations that became accessible, I would conclude that this manuscript goes far beyond previous literature and deserves publication. Not all open and puzzling questions could be answered about the origin of this unexpected impact of the terminal atom on the transport properties of the molecular junction but, in my opinion, it opened a new door and may motivate more studies with other molecules and other type of junctions which will contribute to the consolidation of the present observations and explanations. Therefore, I am convinced that this manuscript will find considerable impact in the scientific community.

Reviewer #5:

Remarks to the Author:

This study is a good piece of work in the field of molecular electronics. The authors indeed demonstrated an efficient method to modulate charge transport in molecular junctions by changing

the tunnelling decay coefficient through terminal-atom substitution. It is nice that the authors varied $X=H, F, Cl, Br, I$ in junctions with $S(CH_2)_{(10-18)}X$, modulated the $S(CH_2)_nX$ //electrode interface, and reduced the contact resistance. However, on the basis of these results, they claimed that they turned insulating saturated wires into molecular conductors, which is intrinsically wrong. The experiments were well performed, but obvious discrepancies exist. I would like to suggest publication in another specific journal after considering major revisions as follows.

1, The experiments the authors demonstrated in this manuscript showed efficient modulation of the $S(CH_2)_nX$ //electrode interface and thus the coupling strength between the molecules and electrodes. Naturally, they reduced the contact resistance. Then, the authors concluded that they changed the conductivity nature of the molecular backbone. This is scientifically not correct. The substitution of the terminal atom of $S(CH_2)_nX$ did strengthen the coupling strength between the molecules and electrodes and increase the molecular conductance, but cannot intrinsically change the nature of the conductance of the molecular backbone. It is natural that the increased coupling strength between the molecules and electrodes leads to the increase of the molecular conductance. This method only realised interface modulation, absolutely not the change of the nature of molecular conductivity. The title and the corresponding descriptions in the manuscript should be revised and clarified.

2, The authors did the good experiments to reduce the value of β by varying $X=H, F, Cl, Br, I$ in junctions with $S(CH_2)_{(10-18)}X$. This strategy is capable of tuning the tunneling rate and the polarizability of the molecules of a SAM, which has been basically reported by the same group (Adv. Mater. 2015, 27, 6689-6695). This fact indeed obviously decreases the novelty of this work.

3, The definition or the explanation of Γ , which I think is coupling strength between the molecules and electrodes. should be provided at its first appearance.

4, In line 24, "shows" should be "show".

5, Please use the uniform form of "characterisation" in the manuscript.

We thank the reviewers for their comments on our manuscript.

Detailed response to the reviewer comments

Reviewer #2 (Remarks to the Author):

Comment 1:

The revised manuscript addresses all of the scientific concerns of the previous versions, but the author's assertion that manipulating Beta values of alkanes is an important breakthrough and departure from earlier work still seems over stated.

This work is very rigorous and the phenomenon is very well characterized, but fundamentally, Beta is proportional to the barrier height, so any change to the barrier height will change Beta. The claim in this work is that the authors are manipulating Beta for alkanes and are doing so by changing a single atom. But that atom is a halogen, which is a significant change. Hexane, 1-chlorohexane and 1-iodohexane have very different physical properties and reactivities. So to some extent the central claim is semantic, since one could just as easily subjectively characterize the addition of a ferrocene as a minor change that has a significant effect on an alkane.

Put another way: *reductio ad absurdum*, the claim is that manipulating Beta in alkanes without any synthetic modification is the underlying motivation. But that can be done by electrostatic gating, which was already demonstrated in 2009 (10.1038/nature08639).

Also, technically, this is not the first demonstration of manipulating Beta in alkanes: 10.1038/s41563-020-00876-2

Granted the effect is smaller, but it is also a single atom exerting a large effect on coupling, just at the opposite electrode. And it most certainly opens up new ways to control tunnelling rates in junctions, since the effect on conductance is significant.

This is definitely interesting work, but the authors should nonetheless consider tempering their language a bit and highlight the magnitude of the effect on Beta, which is an interesting observation, rather than the novelty of the phenomenon itself, which leans on semantics.

Response

We thank the reviewer for supporting publication of our work. We thank the reviewer also for the detailed explanation of their concern regarding potential overstatement of our findings. We understand now how our original statements were misleading. Indeed, our intention was to say that we change the conductance of the *junction*, defined as a unique physical system composed of electrodes and the molecules. Reviewer 5 made a similar point. It is important to consider the junction and NOT the individual components of it separately because, e.g., the effects we reported are largest for the *longest* molecules we study. The current changes by ~ 5 orders of magnitude with changing X in junctions where monolayers are derived from S(CH₂)₁₈X, the longest molecules surveyed in this study, and these changes are much larger than for the shorter molecules we reported. We understand now what caused the unintended confusion, and we have re-phrased the corresponding sentence as the reviewer suggested. We also added the interesting paper as suggested by the reviewer (*Nat. Mater.* **2021**,

10.1038/s41563-020-00876-2; note this work was accepted after our initial submission) although a systematic control over β is not demonstrated in this work.

Changes to the manuscript

On Page 4 Line 90, we added:

“Recently, Chen and co-workers reported a method using bimetallic electrodes to enhance the conductance of $\text{HO}_2\text{C}(\text{CH}_2)_n\text{CO}_2\text{H}$ single-molecule junctions *via* the surface *d*-band.⁴⁰ They improved the interfacial interactions between molecules and transition metal electrodes, promoting interfacial electron transport.”

On Page 6 Line 125, we amended the following sentence from:

“In terms of absolute values of J along long molecular wires of $\text{S}(\text{CH}_2)_{18}\text{X}$, the value of J increases by a factor of $10^{4.5}$ when X is changed from H to I.”

To

“Changing X from H to I in the long $\text{S}(\text{CH}_2)_{18}\text{X}$ molecular wire gives a factor of $10^{\sim 5}$ increase in J . For $\text{S}(\text{CH}_2)_{10}\text{X}$, the currents change by a factor of $10^{\sim 2}$. As we will discuss below, these observations cannot be explained by changes in the molecule–electrode interfaces, or contact resistances, alone.”

On Page 6 Line 129, we changed the sentences from

“To the best of our knowledge, this is the first example of tuning β of alkyl chains opening new ways to control tunnelling rates across junctions.”

to

“While we have shown before that the halide group affects the current and the dielectric constant in $\text{Ag-S}(\text{CH}_2)_{11}\text{X//GaO}_x/\text{EGaIn}$ junctions,⁴⁶ here we demonstrate that the value of β can be controlled by changing X without the need to modify the chemical structure of the molecular backbone. This change in β explains why the largest change in current is found for the longest molecules studied in this work.”

Reviewer #4 (Remarks to the Author):

In my opinion, Xiaoping Chen and co-workers gathered a very broad and detailed collection of experimental and theoretical data utilizing various state-of-the-art investigation techniques to study in a very systematic manner the impact of the terminal group of alkane thiol SAMs on their charge transport characteristics. Considering the large number of different molecule types that were characterized (different chain length and different terminal groups) as well as the extension of the investigation methods in comparison to previous papers and the interpretations that became accessible, I would conclude that this manuscript goes far beyond previous literature and deserves publication. Not all open and puzzling questions could be answered about the origin of this unexpected impact of the terminal atom on the transport properties of the molecular junction but, in my opinion, it opened a new door and may motivate more studies with other molecules and other type of junctions which

will contribute to the consolidation of the present observations and explanations. Therefore, I am convinced that this manuscript will find considerable impact in the scientific community.

Response

We thank the reviewer for the very positive comments on our paper and for supporting publication of our work in Nature Communications. Indeed, we hope that our work will inspire others to investigate the role of polarizable terminal groups in the tunnelling rates in molecular junctions. It will be very interesting to see how such molecules behave in other types of junctions.

Reviewer #5 (Remarks to the Author):

This study is a good piece of work in the field of molecular electronics. The authors indeed demonstrated an efficient method to modulate charge transport in molecular junctions by changing the tunnelling decay coefficient through terminal-atom substitution. It is nice that the authors varied X=H, F, Cl, Br, I in junctions with S(CH₂)_nX, modulated the S(CH₂)_nX//electrode interface, and reduced the contact resistance. However, on the basis of these results, they claimed that they turned insulating saturated wires into molecular conductors, which is intrinsically wrong. The experiments were well performed, but obvious discrepancies exist. I would like to suggest publication in another specific journal after considering major revisions as follows.

Response

We thank the reviewer for acknowledging the importance of our work. We now explain our findings more clearly to avoid any potential for misunderstanding.

Comment 1:

1, The experiments the authors demonstrated in this manuscript showed efficient modulation of the S(CH₂)_nX//electrode interface and thus the coupling strength between the molecules and electrodes. Naturally, they reduced the contact resistance. Then, the authors concluded that they changed the conductivity nature of the molecular backbone. This is scientifically not correct. The substitution of the terminal atom of S(CH₂)_nX did strengthen the coupling strength between the molecules and electrodes and increase the molecular conductance, but cannot intrinsically change the nature of the conductance of the molecular backbone. It is natural that the increased coupling strength between the molecules and electrodes leads to the increase of the molecular conductance. This method only realised interface modulation, absolutely not the change of the nature of molecular conductivity. The title and the corresponding descriptions in the manuscript should be revised and clarified.

Response

Of course the reviewer is correct that contact resistance and coupling strength depend on each other as we have mentioned in our manuscript, but this effect alone cannot explain our results as we discuss in great detail on Pages 17 to 18 (especially Line 399-406). In contrast to the assessment made by the reviewer, in our work we conclude that three factors change β in our case: the tunnelling barrier height, the shape of the potential profile, and the coupling

strength. Hence, our work provides a clear example of how the molecules and the electrodes of the junction form together a unique physical system which typically has completely different properties than what could be inferred from a simple superposition of the properties of the constituent parts or interfaces in isolation. A case in point against the assessment made by the reviewer that we “only realised interface modulation” is that the largest effects in the currents are found by us for the longest molecules: for the $S(CH_2)_{18}X$ derivatives, X has the largest influence on the current which increase by a factor of $10^{\sim 5}$ when X changes from H to I (Page 6, Line 125-129). In contrast, the currents only change by a factor of $10^{\sim 2}$ for $S(CH_2)_{10}X$. This observation is in stark contrast to the reasoning of the reviewer and cannot be explained by “interface modulation” (or contact resistance) which would affect all molecules equally irrespective of their length. Therefore, the interfaces alone cannot explain our observed changes in β as explained on Pages 17 to 18.

We would like to point the Reviewer also to the work of the Frisbie group (*J. Am. Chem. Soc.* **2019**, 141, 8, 3670–3681; *J. Am. Chem. Soc.* **2019**, 141, 45, 18182–18192) who have shown that changing the molecule–electrode coupling strength Γ does change the conductivity of the molecules, but not the β values. Their work reinforces our conclusion that interfaces alone cannot explain our observations.

As detailed also in the response to Reviewer 2, we now clearly state that we change the conductivity of the *junction* to avoid potential misunderstanding.

Changes to the manuscript

Please see our response to Reviewer 2. In addition:

On Page 10, Line 231, we added:

“Frisbie and co-workers have shown that a decrease in R_C by increasing the work function of the bare metal electrodes increases the conductivity of molecular junctions (with H or S terminal atoms), which was mainly driven by a large increase in Γ , with changes in δE_{ME} and β playing only a minor role.^{9,38}”

To clarify ourselves, we added the following sentence to the Discussion section, on Page 17, Line 394:

“The largest effects of X on J are found in the longest molecules of $S(CH_2)_{18}X$ where the J increases by a factor of $10^{\sim 5}$ when X changes from H to I. In contrast, for the shortest molecule $S(CH_2)_{10}X$, J increases by a factor of $10^{\sim 2}$, indicating that the observed changes in J , and the corresponding values of β , are driven by more than just changes in the interfaces, which has been discussed previously by Frisbie and co-workers^{9,38}.”

Comment 2:

2, The authors did the good experiments to reduce the value of β by varying X=H, F, Cl, Br, I in junctions with $S(CH_2)_{(10-18)}X$. This strategy is capable of tuning the tunneling rate and the polarizability of the molecules of a SAM, which has been basically reported by the same group (*Adv. Mater.* 2015, 27, 6689-6695). This fact indeed obviously decreases the novelty of this work.

Response

The strategy of tuning tunnelling rates by X substitution was not part of our earlier study. Our earlier work only reported changes in the observed currents as a function of X for molecules with a fixed molecular length, i.e., S(CH₂)₁₁X. Obviously, the tunnelling decay coefficient cannot be determined in such an experiment. In this work, we changed the length of the molecular backbone allowing us to determine that the nature of X changes β and discussed the interrelationship between β and dielectric constant. This is the novelty of the present work.

We believe we have altered our manuscript during the previous rounds of review to properly reflect what is novel about our present contribution to the satisfaction of Reviewer 2 (among other reviewers) as described above (see, e.g., Page 6 Line 129-133 of the manuscript, where we added the sentence to mention the difference between (Adv. Mater. 2015, 27, 6689-6695) and this work). In response to Reviewer 2, we have now fine-tuned our claims regarding novelty.

Changes to the manuscript

Please see our response to Reviewer 2

Comment 3:

3, The definition or the explanation of Γ , which I think is coupling strength between the molecules and electrodes. should be provided at its first appearance.

Response

We thank the referee for catching this oversight, and we now define “the coupling strength between the molecules and electrodes (Γ)” at its first appearance on Page 4, Line 71.

Comment 4:

4, In line 24, “shows” should be “show”.

Response

Thank you for alerting us to this mistake which we have corrected.

Comment 5:

5, Please use the uniform form of “characterisation” in the manuscript.

Response

We have corrected this oversight.

Editorial changes:

The following are editorial changes according to the Nature Communications Formatting Instructions. The editorial requests are resolved without scientific changes to the manuscript.

- (1) We have updated the editorial policy checklist.

- (2) The Data and code availability are already in the last version of manuscript. We have added the statement “Source data are provided with this paper” to the “Data availability” Section. An Excel file containing all the raw data of the figures is also uploaded to the “Online submission system”.
- (3) All corresponding authors have linked their ORCID account on the Manuscript Tracking System.

Reviewers' Comments:

Reviewer #2:

Remarks to the Author:

The claims of the paper are much clearer in this version. It is suitable for publication in its current form.

Reviewer #5:

Remarks to the Author:

I went through the authors' responses with interest. Unfortunately, they did not answer my scientific concerns. Considering the limited scientific contribution and novelty although the authors did perform the systematic and nice characterization, I think it is more suitable for another specific journal.

We thank the reviewers for their comments on our manuscript.

Detailed response to the reviewer comments

Reviewer #2 (Remarks to the Author):

Comment 1:

The claims of the paper are much clearer in this version. It is suitable for publication in its current form.

Response: we thank the reviewer for supporting our publication on Nature Communications.

Changes to the manuscript:

No changes.

Reviewer #5 (Remarks to the Author):

I went through the authors' responses with interest. Unfortunately, they did not answer my scientific concerns. Considering the limited scientific contribution and novelty although the authors did perform the systematic and nice characterization, I think it is more suitable for another specific journal.

Response: we thank the reviewer for going through the response letter and manuscript again. In contrast to Reviewer 2 who has seen the work again, and the other Reviewers, Reviewer 5 still questions novelty but without giving us a specific reason. We strongly feel that in the previous round of review that we have explained in sufficient detail, while avoiding over-hyping, the new insights our work brings as recognized by the other reviewers.

Changes to the manuscript:

No changes.

Editorial requests and formatting changes:

SUBMISSION INFORMATION

In order to accept your paper, we require the following:

- A revised author checklist describing your response to our editorial requests (attached).

Response: we have updated the author checklist and responded to all the editorial requests, the modifications of the formats according to the "author checklist" are highlighted in yellow in the manuscript and supplementary information.

The main changes are:

- 1) We defined all the abbreviations and symbols in figures legends.
- 2) All the errors in figures legends are defined.
- 3) We formatted the Table 1 in manuscript.
- 4) We added experimental details in supplementary information where citations to previously papers were used.

On Page S24, Line 458, we added the following sentences:

"Briefly, 200 nm Ag (with a purity of 99.99 %, purchased from ACI Alloy, USA) was deposited onto a clean 6 inch Prime Si (100) wafer (SYST Integration Pte Ltd,

Singapore) under vacuum level of 5×10^{-5} Pa using a thermal evaporator (DZ270, SKY Technology Development Co., Ltd, Shenyang, China). The deposition rates were ~ 0.3 Å/s for the first 20 nm and ~ 1.0 Å/s for the remaining thickness. Piranha solution cleaned glass slides with dimensions of 1×2 cm were glued to the Ag surface using a thermal adhesive (EPOTEK 353ND, purchased from EPOXY TECHNOLOGY, INC. Massachusetts, USA), which was then cured at 80 °C using an oven (ZRD-A5110A, Zhicheng Inc. Shanghai, China) for 16 hours. The Ag substrates were stored in a clean dry box and the substrates were template-stripped just before use.”

On Page S24, Line 480, we added the following sentences:

“Briefly, d_{SAM} was calculated using the effective intensities I_{90° and I_{40° of the S $2p$ spectra (at the peak centre of 162 eV), as shown in Eq. S1, d_1 ($=1.5$ Å) is estimated from the radius of the S atoms and the S-C bond. To this d , Ag-S distance d_{Ag-S} of 1.8 Å was added to get d_{SAM} .

$$d_{SAM} = \frac{\lambda \sin 90^\circ \sin 40^\circ \left[\ln \left(\frac{I_{90^\circ}}{I_{40^\circ}} \right) + \ln \left(1 - e^{-\frac{d_1}{\lambda \sin 40^\circ}} \right) - \ln \left(1 - e^{-\frac{d_1}{\lambda \sin 90^\circ}} \right) \right]}{\sin 90^\circ - \sin 40^\circ} + d_{Ag-S} \quad (S1)''$$

On Page S30, Line 530, we added:

“Briefly, the bias was applied to the junction following the sequence of $0 \rightarrow 0.5 \rightarrow 0 \rightarrow -0.5 \rightarrow 0$ V for each $J(V)$ trace.”

- A separate point-by-point response to the reviewers’ comments, reproduced verbatim.

Response: we have submitted a separate point-by-point response to the reviewers’ comments in the file “**Response to referees**”.

- The final version of your manuscript as a Word or LaTeX file, with all changes highlighted in the text and any tables prepared using the table menu in Word or the table environment in LaTeX.

Response: our manuscript is in a Word format and all changes are highlighted in yellow.

- If using LaTeX, please use numerical references only for citations, and include the references within the manuscript file itself. If you wish to use BibTeX, please copy the reference list from the .bbl file, paste it into the main manuscript .tex file, and delete the associated \bibliography and \bibliographystyle commands.

Response: our manuscript is in a Word file.

- The complete author list provided in the manuscript file, which must match that given on our manuscript tracking system. The author list in the main manuscript file will be used during typesetting of your article.

Response: the author list provided in the manuscript files matches the one in the tracking system.

- Production-quality versions of each figure as a separate file containing all panels. To ensure the swift processing of your paper, please provide the highest quality versions of your images and when combining different figure parts into one file for layout, use a vector-based application such as Adobe Illustrator or Microsoft Powerpoint. We recommend .ai, .eps, .pdf, .ppt. Figures divided into panels should be labelled with a lower-case, boldface 'a', 'b', etc. in the top left-hand corner. If resolution is not of sufficient quality, production of your paper

will be held whilst replacement files are obtained. For detailed guidance on figure preparation, see <https://www.nature.com/documents/aj-artworkguidelines.pdf>
Response: we provide production-quality versions of each figure in a Microsoft Powerpoint file. Figures are labelled with a lower-case in the top left-hand corner.

- Please note that we do not modify the text in figures to conform to style during the production process. Please ensure that your figures are presented accurately and adhere to the guidance provided.

Response: we have checked the figures according to the guidance.

- Any updated checklists that verify compliance with our research ethics and data reporting standards in PDF format.

Response: we have submitted the updated file “nr-editorial-policy-checklist”.

- The final version of the Supplementary Information in one PDF file.

Response: we provide the final version of Supplementary Information in one PDF file.

- Any Supplementary Movie, Audio, Data and Software submitted as separate files. Supplementary Data and Source Data must be provided as .xls, .xlsx or .zip files, while Supplementary Software must be supplied as .zip files..

** Please note that we do not edit Supplementary Information files; they must be finalised prior to acceptance of the paper. **

Response: we provide the Supplementary Data in separate zip files and the Source Data are provided in .xlsx files.

- If you wish, an interesting image (but not an illustration or schematic) for consideration as a Featured Image on the Nature Communications homepage. The file should be 1200x675 pixels in RGB format and should be uploaded as a Related Manuscript File. In addition to our home page, we may also use this image (with credit) in other journal-specific promotional material.

Response: thanks for the invitation, but we may not be able to submit a Featured Image at this stage.

- Completed and signed copies of our Multimedia License to Publish (LTP) for any Featured Image suggestions (please use one form for each image and give a scientific description of the image in the 'title' field; do not use "Featured Image" as a

title): <http://www.nature.com/documents/sn1-multimedia-ltp.docx>

Response: NA

OPEN ACCESS

Nature Communications is a fully open access journal. Articles are made freely accessible on publication under a [CC BY license](http://creativecommons.org/licenses/by/4.0) (Creative Commons Attribution 4.0 International License). This license allows maximum dissemination and re-use of open access materials and is preferred by many research funding bodies.

For further information about article processing charges, open access funding, and advice and support from Nature Research, please visit <http://www.nature.com/ncomms/about/open-access>

At acceptance, the corresponding author will be required to complete an Open Access Licence to Publish on behalf of all authors, declare that all required third party permissions have been obtained and provide billing information in order to pay the article-processing charge (APC) via credit card or invoice.

Please note that your paper cannot be sent for typesetting to our production team until we have received these pieces of information; **therefore, please ensure that you have this information ready when submitting the final version of your manuscript.**

Response: yes, we have the third party permissions and billing information ready.

ORCID

Nature Communications is committed to improving transparency in authorship. As part of our efforts in this direction, we are now requesting that all authors identified as ‘corresponding author’ create and link their Open Researcher and Contributor Identifier (ORCID) with their account on the Manuscript Tracking System (MTS) prior to acceptance. ORCID helps the scientific community achieve unambiguous attribution of all scholarly contributions. For more information please visit <http://www.springernature.com/orcid> For all corresponding authors listed on the manuscript, please follow the instructions in the link below to link your ORCID to your account on our MTS before submitting the final version of the manuscript. If you do not yet have an ORCID you will be able to create one in minutes.

Response: yes, we have all the corresponding authors linked to the MTS.

POLICIES

If you opted into the journal hosting details of a preprint version of your manuscript via a link on our dedicated website (<https://nature-research-under-consideration.nature.com>), it will remain on this site while you are revising your manuscript. If you wish to remove these details, please email naturecommunications@nature.com indicating your manuscript number and the link on our website that was previously sent to you. For more information, please refer to our FAQ page at <https://nature-research-under-consideration.nature.com/posts/19641-frequently-asked-questions>

Response: it is fine to keep the preprint version on website.

In recognition of the time and expertise our reviewers provide to Nature Communications’s editorial process, as of November 2018, we formally acknowledge their contribution to the external peer review of articles published in the journal. All peer-reviewed content will carry an anonymous statement of peer reviewer acknowledgement, and for those reviewers who give their consent, we will publish their names alongside the published article. For more information, please refer to our FAQ page at <https://www.nature.com/documents/ncomms-reviewer-information.pdf>

Response: NA

Nature Research journals encourage authors to share their step-by-step experimental protocols on a protocol sharing platform of their choice. Where such protocols are available, please provide a DOI or other citation details in the paper. Nature Research’s *Protocol Exchange* is a free-to-use and open resource for protocols; protocols deposited in *Protocol Exchange* are citable and can be linked from the published article. More details can found at <https://www.nature.com/protocolexchange/about>

Response: the step-by-step experimental protocols are described in detail with citations in the Supporting Information file.